# ON INCORPORATING INDUCTIVE BIASES INTO VAES

**Ning Miao**[1*]  **Emile Mathieu**[1]  **N. Siddharth**[2]  **Yee Whye Teh**[1]  **Tom Rainforth**[1*]

## ABSTRACT

We explain why directly changing the prior can be a surprisingly ineffective mechanism for incorporating inductive biases into variational auto-encoders (VAEs), and introduce a simple and effective alternative approach: *Intermediary Latent Space VAEs* (InteL-VAEs). InteL-VAEs use an intermediary set of latent variables to control the stochasticity of the encoding process, before mapping these in turn to the latent representation using a parametric function that encapsulates our desired inductive bias(es). This allows us to impose properties like sparsity or clustering on learned representations, and incorporate human knowledge into the generative model. Whereas changing the prior only indirectly encourages behavior through regularizing the encoder, InteL-VAEs are able to directly enforce desired characteristics. Moreover, they bypass the computation and encoder design issues caused by non-Gaussian priors, while allowing for additional flexibility through training of the parametric mapping function. We show that these advantages, in turn, lead to both better generative models and better representations being learned.

## 1 INTRODUCTION

VAEs provide a rich class of deep generative models (DGMs) with many variants (Kingma & Welling, 2014; Rezende & Mohamed, 2015; Burda et al., 2016; Gulrajani et al., 2016; Vahdat & Kautz, 2020). Based on an encoder-decoder structure, VAEs encode datapoints into latent embeddings before decoding them back to data space. By parameterizing the encoder and decoder using expressive neural networks, VAEs provide a powerful basis for learning both generative models and representations.

The standard VAE framework assumes an isotropic Gaussian prior. However, this can cause issues, such as when one desires the learned representations to exhibit some properties of interest, for example sparsity (Tonolini et al., 2020) or clustering (Dilokthanakul et al., 2016), or when the data distribution has very different topological properties from a Gaussian, for example multi-modality (Shi et al., 2020) or group structure (Falorsi et al., 2018). Therefore, a variety of recent works have looked to use non-Gaussian priors (van den Oord et al., 2017; Tomczak & Welling, 2018; Casale et al., 2018; Razavi et al., 2019; Bauer & Mnih, 2019), often with the motivation of adding inductive biases into the model (Davidson et al., 2018b; Mathieu et al., 2019b; Nagano et al., 2019; Skopek et al., 2019).

In this work, we argue that this approach of using non-Gaussian priors can be a problematic, and even ineffective, mechanism for adding *inductive biases* into VAEs. Firstly, non-Gaussian priors will often necessitate complex encoder models to maintain consistency with the prior's shape and dependency structure (Webb et al., 2018), which typically no longer permit simple parameterization. Secondly, the latent encodings are still not guaranteed to follow the desired structure because the 'prior' only appears in the training objective as a regularizer on the encoder. Indeed, Mathieu et al. (2019b) find that changing the prior is typically insufficient in practice to learn the desired representations at a *population level*, with mismatches occurring between the data distribution and learned model.

To provide an alternative, more effective, approach that does not suffer from these pathologies, we introduce *Intermediary Latent Space VAEs* (InteL-VAEs), an extension to the standard VAE framework that allows a wide range of powerful inductive biases to be incorporated while maintaining an isotropic Gaussian prior. This is achieved by introducing an *intermediary* set of latent variables that deal with the stochasticity of the encoding process *before* incorporating the desired inductive biases via a parametric function that maps these intermediary latents to the latent representation itself, with the decoder taking this final representation as input. See Fig. 1 for an example.

---

[1]Department of Statistics, University of Oxford,  [2]University of Edinburgh
[*]Correspondence to: Ning Miao <ning.miao@stats.ox.ac.uk>, Tom Rainforth <rainforth@stats.ox.ac.uk>

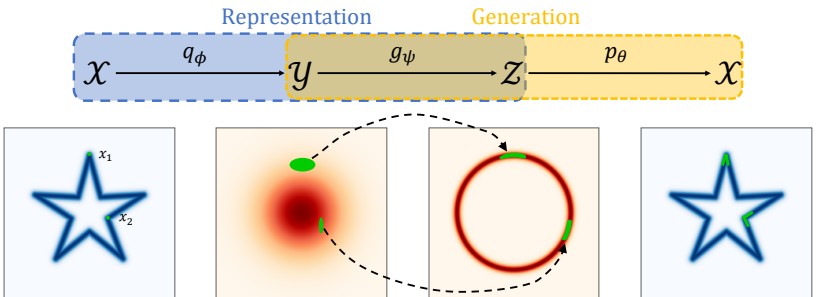

Figure 1: Example InteL-VAE with star-like data. We consider the auto-encoding for two example datapoints ($x_1$ and $x_2$, shown in green), which are first stochastically mapped to $\mathcal{Y}$ using a Gaussian encoder. This embedding is then pushed forward to $\mathcal{Z}$ using the *non-stochastic* mapping $g_\psi$, which is a radial mapping to enforce a spherical distribution. Decoding is then done in the standard way from $\mathcal{Z}$, with the complexity of the decoder mapping simplified by the induced structural properties of $\mathcal{Z}$.

The InteL-VAE framework provides a variety of advantages over directly replacing the prior. Firstly, it directly enforces our inductive biases on the representations, rather than relying on the regularizing effect of the prior to encourage this implicitly. Secondly, it provides a natural congruence between the generative and representational models via sharing of the mapping function, side-stepping the issues that non-Gaussian priors can cause for the inference model. Finally, it allows for more general and more flexible inductive biases to be incorporated, by removing the need to express them with an explicit density function and allowing for parts of the mapping to be learned during training.

To further introduce a number of novel specific realizations of the InteL-VAE framework, showing how they can be used to incorporate various inductive biases, enforcing latent representations that are, for example, multiply connected, multi-modal, sparse, or hierarchical. Experimental results show their superiority compared with baseline methods in both generation and feature quality, most notably providing state-of-the-art performance for learning sparse representations in the VAE framework.

To summarize, we a) highlight the need for inductive biases in VAEs and explain why directly changing the prior is a suboptimal means for incorporating them; b) propose InteL-VAEs as a simple but effective general framework to introduce inductive biases; and c) introduce specific InteL-VAE variants which can learn improved generative models and representations over existing baselines on a number of tasks. Accompanying code is provided at `https://github.com/NingMiao/InteL-VAE`.

## 2 THE NEED FOR INDUCTIVE BIASES IN VAEs

Variational auto-encoders (VAEs) are deep stochastic auto-encoders that can be used for learning both deep generative models and low-dimensional representations of complex data. Their key components are an encoder, $q_\phi(z|x)$, which probabilistically maps from data $x \in \mathcal{X}$ to latents $z \in \mathcal{Z}$; a decoder, $p_\theta(x|z)$, which probabilistically maps from latents to data; and a prior, $p(z)$, that completes the generative model, $p_\theta(z)p_\theta(x|z)$, and regularizes the encoder during training. The encoder and decoder are parameterized by deep neural networks and are simultaneously trained using a dataset $\{x_1, x_2, ..., x_N\}$ and a variational lower bound on the log-likelihood, most commonly,

$$\mathcal{L}(x, \theta, \phi) := \mathbb{E}_{z \sim q_\phi(z|x)} \left[ \log p_\theta(x|z) \right] - D_{\text{KL}} \left( q_\phi(z|x) \parallel p(z) \right). \tag{1}$$

Namely, we optimize $\mathcal{L}(\theta, \phi) := \mathbb{E}_{x \sim p_{\text{data}}(x)} \left[ \mathcal{L}(x, \theta, \phi) \right]$, where $p_{\text{data}}(x)$ represents the empirical data distribution. Here the prior is typically fixed to a standard Gaussian, i.e. $p(z) = \mathcal{N}(z; 0, I)$.

While it is well documented that this standard VAE setup with a 'Gaussian' latent space can be suboptimal (Davidson et al., 2018a; Mathieu et al., 2019b; Tomczak & Welling, 2018; Bauer & Mnih, 2019; Tonolini et al., 2020), there is perhaps less of a unified high-level view on exactly when, why, and how one should change it to incorporate inductive biases. Note here that the prior does not play the same role as in a Bayesian model: because the latents themselves are somewhat arbitrary and the model is learned from data, it does not encapsulate our initial beliefs in the way one might expect.

We argue that there are two core reasons why inductive biases can be important for VAEs: (a) standard VAEs can fail to encourage, and even prohibit, desired structure in the *representations* we learn; and (b) standard VAEs do not allow one to impart prior information or desired topological characteristic into the *generative model*.

Considering the former, one often has some a priori desired characteristics, or constraints, on the representations learned (Bengio et al., 2013). For example, sparse features can be desirable because they can improve data efficiency (Yip & Sussman, 1997), and provide robustness to noise (Wright et al., 2009; Ahmad & Scheinkman, 2019) and attacks (Gopalakrishnan et al., 2018). In other settings one might desire clustered (Jiang et al., 2017), disentangled (Ansari & Soh, 2019; Kim & Mnih, 2018; Higgins et al., 2018) or hierarchical representations (Song & Li, 2013; Sønderby et al., 2016; Zhao et al., 2017). The KL-divergence term in Eq. (1) regularizes the encoding distribution towards the prior and, as a standard Gaussian distribution typically does not exhibit our desired characteristics, this regularization can significantly hinder our ability to learn representations with the desired properties.

Not only can this be problematic at an individual sample level, it can cause even more pronounced issues at the *population level*: desired structural characteristics of our representations often relate to the pushforward distribution of the data in the latent space, $q_\phi(z) := \mathbb{E}_{p_{\text{data}}(x)}[q_\phi(z|x)]$, which is both difficult to control and only implicitly regularized to the prior (Hoffman & Johnson, 2016).

Inductive biases can also be essential to the generation quality of VAEs: because the generation process of standard VAEs is essentially pushing-forward the Gaussian prior on $\mathcal{Z}$ to data space $\mathcal{X}$ by a 'smooth' decoder, there is an underlying inductive bias that standard VAEs prefer sample distributions with similar topology structures to Gaussians. As a result, VAEs can perform poorly when the data manifold exhibits certain different topological properties (Caterini et al., 2020). For example, they can struggle when data is clustered into unconnected com-

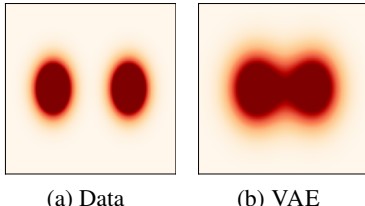

(a) Data      (b) VAE

Figure 2: VAE learned generative distribution $\mathbb{E}_{p(z)}[p_\theta(x|z)]$ for mixture data.

ponents as shown in Fig. 2, or when data is not simply-connected. This renders learning effective mappings using finite datasets and conventional architectures (potentially prohibitively) difficult. In particular, it can necessitate large Lipschitz constants in the decoder, causing knock-on issues like unstable training and brittle models (Scaman & Virmaux, 2018), as well as posterior collapse (van den Oord et al., 2017; Alemi et al., 2018). In short, the Gaussian prior of a standard VAE can induce fundamental topological differences to the true data distribution (Falorsi et al., 2018; Shi et al., 2020).

## 3 SHORTFALLS OF VAES WITH NON-GAUSSIAN PRIORS

Though directly replacing the Gaussian prior with a different prior sounds like a simple solution, effectively introducing inductive biases can, unfortunately, be more complicated.

Firstly, the only influence of the prior during training is as a regularizer on the encoder through the $D_{\text{KL}}(q_\phi(z|x) \| p(z))$ term. This regularization is always competing with the need for effective reconstructions and only has an indirect influence on $q_\phi(z)$. As such, simply replacing the prior can be an ineffective way of inducing desired structure at the population level (Mathieu et al., 2019b), particularly if $p(z)$ is a complex distribution that it is difficult to fit (see, e.g., Fig. 3a). Mismatches between $q_\phi(z)$ and $p(z)$ can also have further deleterious effects on the learned generative model: the

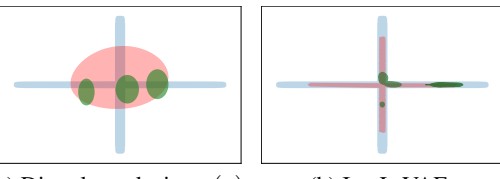

(a) Directly replacing $p(z)$      (b) InteL-VAE

Figure 3: Prior-encoder mismatch. We train (a) a VAE with a sparse prior and (b) an InteL-VAE with a sparse inductive bias on 2 dimensional sparse data. Figure shows target latent distribution $p(z)$ (blue), learned variational embeddings $q_\phi(z|x)$ of exemplar data (green), and data pushforward $q_\phi(z)$ (red shadow) for each method. Simply replacing the prior does not help the VAE match prior structure on either a per-sample or population level, whereas InteL-VAE produces an effective match.

former represents the distribution of the data in latent space during training, while the latter is what is used by the learned generative model, leading to unrepresentative generations if there is mismatch.

Secondly, it can be extremely difficult to construct appropriate encoder mappings and distributions for non-Gaussian priors. While the typical choice of a mean-field Gaussian for the encoder distribution is simple, easy to train, and often effective for Gaussian priors, it is often inappropriate for other choices of prior. For example, in Fig. 3, we consider replacement with a sparse prior. A VAE with a Gaussian encoder struggles to encode points in a manner that even remotely matches the prior. One

might suggest replacing the encoder distribution as well, but this has its own issues, most notably that other distributions can be hard to effectively parameterize or train. In particular, the form of the required encoding noise might become heavily spatially variant; in our sparse example, the noise must be elongated in a particular direction depending on where the mean embedding is. If the prior has constraints or topological properties distinct from the data, it can even be difficult to learn a mean encoder mapping that respects these, due to the continuous nature of neural networks.

## 4   THE INTEL-VAE FRAMEWORK

To solve the issues highlighted in the previous section, and provide a principled and effective method for adding inductive biases to VAEs, we propose *Intermediary Latent Space VAEs* (InteL-VAEs). The key idea behind InteL-VAEs is to introduce an *intermediary* set of latent variables $y \in \mathcal{Y}$, used as a stepping stone in the construction of the *representation $z \in \mathcal{Z}$*. Data is initially encoded in $\mathcal{Y}$ using a conventional VAE encoder (e.g. a mean-field Gaussian) before being passed through a *non-stochastic* mapping $g_\psi : \mathcal{Y} \mapsto \mathcal{Z}$ that incorporates our desired inductive biases and which can be trained, if needed, through its parameters $\psi$. The prior is defined on $\mathcal{Y}$ and taken to be a standard Gaussian, $p(y) = \mathcal{N}(y; 0, I)$, while our representations, $z = g_\psi(y)$, correspond to a pushforward of $y$. By first encoding datapoints to $y$, rather than $z$ directly, we can deal with all the encoder and prior stochasticity in this first, well-behaved, latent space, while maintaining $z$ as our representation and using it for the decoder $p_\theta(x|z)$. In principle, $g_\psi$ can be any arbitrary parametric (or fixed) mapping, including non-differentiable or even discontinuous functions. However, to allow for reparameterized gradient estimators (Kingma & Welling, 2014; Rezende & Mohamed, 2015), we will restrict ourselves to $g_\psi$ that are sub-differentiable (and thus continuous) with respect to both their inputs and parameters. Note that setting $g_\psi$ to the identity mapping recovers a conventional VAE.

As shown in Fig. 1, the auto-encoding process is now $\mathcal{X} \xrightarrow{q_\phi} \mathcal{Y} \xrightarrow{g_\psi} \mathcal{Z} \xrightarrow{p_\theta} \mathcal{X}$. This three-step process no longer unambiguously fits into the encoder-decoder terminology of the standard VAE and permits a variety of interpretations; for now we take the convention of calling $q_\phi(y|x)$ the encoder and $p_\theta(x|z)$ the decoder, but also discuss some alternative interpretations below. We emphasize here that these no longer respectively match up with our representation model—which corresponds to passing an input into the encoder and then mapping the resulting encoding using $g_\psi$—and our generative model—which corresponds to $\mathcal{N}(y; 0, I)p_\theta(x|z = g_\psi(y))$, such that we sample a $y$ from the prior and then pass this through through $g_\psi$ and the decoder in turn.

The mapping $g_\psi$ introduces inductive biases into *both* the generative model and our representations by imposing a particular form on $z$, such as the spherical structure enforced in Fig. 1 (see also Sec. 6). It can be viewed as a *shared module* between them, ensuring congruence between the two. This congruence allows us to more directly introduce inductive biases through careful construction of $g_\psi$, without complicating the process of learning an effective inference network. In particular, because $\mathcal{Y}$ is treated as our latent space for the purposes of training, we sidestep the inference issues that non-Gaussian priors usually cause. Moreover, because all samples must explicitly pass through $g_\psi$ during both training and generation, we can more directly ensure the desired structure is enforced without causing a mismatch in the latent distribution between training and deployment.

**Training**   As with standard VAEs, training of an InteL-VAE is done by maximizing a variational lower bound (ELBO) on the log evidence, which we denote $\mathcal{L}_\mathcal{Y}$. Most simply, we have

$$
\begin{aligned}
\log p_{\theta,\psi}(x) &:= \log \left( \mathbb{E}_{p(y)} \left[ p_\theta(x|g_\psi(y)) \right] \right) = \log \left( \mathbb{E}_{q_\phi(y|x)} \left[ \frac{p_\theta(x|g_\psi(y))\mathcal{N}(y; 0, I)}{q_\phi(y|x)} \right] \right) \\
&\geq \mathbb{E}_{q_\phi(y|x)} [\log p_\theta(x|g_\psi(y))] - D_{\mathrm{KL}} \left( q_\phi(y|x) \,\|\, \mathcal{N}(y; 0, I) \right) =: \mathcal{L}_\mathcal{Y}(x, \theta, \phi, \psi).
\end{aligned}
\tag{2}
$$

Note that the regularization is on $y$, but our representation corresponds to $z = g_\psi(y)$. Training corresponds to the optimization $\arg\max_{\theta,\phi,\psi} \mathbb{E}_{x\sim\mathrm{P_{data}}(x)} [\mathcal{L}_\mathcal{Y}(x, \theta, \phi, \psi)]$, which can be performed using stochastic gradient ascent with reparameterized gradients in the standard manner. Although inductive biases are introduced, the calculation, and optimization, of $\mathcal{L}_\mathcal{Y}$ is thus equivalent to the standard ELBO. In particular, parameterizing $q_\phi(y|x)$ with a Gaussian distribution still yields an analytical $D_{\mathrm{KL}} \left( q_\phi(y|x) \,\|\, \mathcal{N}(y; 0, I) \right)$ term.

**Alternative Interpretations**   It is interesting to note that our representation, $g_\psi(y)$, only appears in the context of the decoder in this training objective. As such, we see that an important alternative interpretation of InteL-VAEs is to consider $g_\psi$ as being a customized first layer in the decoder, and our

test–time representations as partial decodings of the latents $y$. This viewpoint allows it to be applied with more general bounds and VAE variants (e.g. Burda et al. (2016); Le et al. (2018); Maddison et al. (2017); Naesseth et al. (2018); Zhao et al. (2019)), as it requires only a carefully customized decoder architecture during training and an adjusted mechanism for constructing representations at test–time.

Yet another interpretation is to think about InteL-VAEs as implicitly defining a conventional VAE with latents $z$, but where both the non-Gaussian prior, $p_\psi(z)$, and our encoder distribution, $q_{\phi,\psi}(z|x)$, are themselves defined implicitly as pushforwards along $g_\psi$, which acts as a shared module that instills a natural compatibility between the two. Formally we have the following theorem.

**Theorem 1.** *Let $p_\psi(z)$ and $q_{\phi,\psi}(z|x)$ represent the respective pushforward distributions of $\mathcal{N}(0, I)$ and $q_\phi(y|x)$ induced by the mapping $g_\psi : \mathcal{Y} \mapsto \mathcal{Z}$. The following holds for all measurable $g_\psi$:*

$$D_{KL}\left(q_{\phi,\psi}(z|x) \parallel p_\psi(z)\right) \leq D_{KL}\left(q_\phi(y|x) \parallel \mathcal{N}(y; 0, I)\right). \tag{3}$$

*If $g_\psi$ is also an invertible function then the above becomes an equality and $\mathcal{L}_\mathcal{Y}$ equals the standard ELBO on the space of $\mathcal{Z}$ as follows*

$$\mathcal{L}_\mathcal{Y}(x, \theta, \phi, \psi) = \mathbb{E}_{q_{\phi,\psi}(z|x)}[\log p_\theta(x|z)] - D_{KL}\left(q_{\phi,\psi}(z|x) \parallel p_\psi(z)\right). \tag{4}$$

The proof is given in Appendix A. Here, (3) shows that the divergence in our representation space $\mathcal{Z}$ is never more than that in $\mathcal{Y}$, or equivalently that the implied ELBO on the space of $\mathcal{Z}$ is always at least as tight as that on $\mathcal{Y}$; (4) shows they are exactly equal if $g_\psi$ is invertible. As the magnitude of $D_{KL}\left(q_\phi(y|x) \parallel \mathcal{N}(y; 0, I)\right)$ in an InteL-VAE will remain comparable to the KL divergence in a standard Gaussian prior VAE setup, this, in turn, ensures that $D_{KL}\left(q_{\phi,\psi}(z|x) \parallel p_\psi(z)\right)$ does not become overly large. This is in stark contrast to the conventional non-Gaussian prior setup, where it can be difficult to avoid $D_{KL}\left(q_\phi(z|x) \parallel p_\psi(z)\right)$ exploding without undermining reconstruction (Mathieu et al., 2019b). The intuition here is that having the stochasticity in the encoder *before* it is passed through $g_\psi$ ensures that the form of the noise in the embedding is inherently appropriate for the space: the same mapping is used to warp this noise as to define the generative model in the first place. For example, when $g_\psi$ is a sparse mapping, the Gaussian noise in $q_\phi(y|x)$ will be compressed to a sparse subspace by $g_\psi$, leading to a sparse variational posterior $q_{\phi,\psi}(z|x)$ as shown in Fig. 3b. In particular, $q_\phi(y|x)$ does not need to learn any complex spatial variations that result from properties of $\mathcal{Z}$. In turn, InteL-VAEs further alleviate issues of mismatch between $p_\psi(z)$ and $q_{\phi,\psi}(z)$.

**Further Benefits** A key benefit of InteL-VAEs is that the extracted features are *guaranteed* to have the desired structure. Take the spherical case for example, all extracted features $g_\psi(\mu_\phi(x))$ lie within a small neighborhood of the unit sphere. By comparison, methods based on training loss modifications, e.g. Mathieu et al. (2019b), often fail to generate features with the targeted properties.

A more subtle advantage is that we do not need to explicitly specify $p_\psi(z)$. This can be extremely helpful when we want to specify complex inductive biases: designing a non-stochastic mapping is typically much easier than a density function, particularly for complex spaces. Further, this can make it much easier to parameterize and learn aspects of $p_\psi(z)$ in a data-driven manner (see e.g. Sec. 6.3).

## 5 RELATED WORK

**Inductive biases** There is much prior work on introducing human knowledge to deep learning models by structural design, such as CNNs (LeCun et al., 1989), RNNs (Hochreiter & Schmidhuber, 1997) and transformers (Vaswani et al., 2017). However, most of these designs are on the *sample* level, utilizing low–level information such as transformation invariances or internal correlations in each sample. By contrast, InteL-VAEs provide a convenient way to incorporate *population* level knowledge—information about the global properties of data distributions can be effectively utilized.

**Non-Gaussian priors** There is an abundance of prior work utilizing non-Gaussian priors to improve the fit and generation capabilities of VAEs, including MoG priors (Dilokthanakul et al., 2016; Shi et al., 2020), sparse priors (Mathieu et al., 2019b; Tonolini et al., 2020; Barello et al., 2018), Gaussian-process priors (Casale et al., 2018) and autoregressive priors (Razavi et al., 2019; van den Oord et al., 2017). However, these methods often require specialized algorithms to train and are primarily applicable only to specific kinds of data. Moreover, as we have explained, changing the prior alone often provides insufficient pressure on its own to induce the desired characteristics. Others have proposed non-Gaussian priors to reduce the prior-posterior gap, such as Vamp-VAE (Tomczak & Welling, 2018) and LARS (Bauer & Mnih, 2019), but these are tangential to our inductive bias aims.

**Non-Euclidean latents** A related line of work has focused on non-Euclidean latent spaces. For instance Davidson et al. (2018a) leveraged a von Mises-Fisher distribution on a hyperspherical latent space, Falorsi et al. (2018) endowed the latent space with a SO(3) group structure, and Mathieu et al. (2019a); Ovinnikov (2019); Nagano et al. (2019) with hyperbolic geometry. Other spaces like product of constant curvature spaces (Skopek et al., 2019) and embedded manifolds (Rey et al., 2019) have also been considered. However, these works generally require careful design and training.

**Normalizing flows** Our use of a non-stochastic mapping shares some interesting links to normalizing flows (NFs) (Rezende & Mohamed, 2015; Papamakarios et al., 2019; Grathwohl et al., 2018; Dinh et al., 2017; Huang et al., 2018; Papamakarios et al., 2018). Indeed a NF would be a valid choice for $g_\psi$, albeit an unlikely one due to their architectural constraints. However, unlike previous use of NFs in VAEs, our $g_\psi$ is crucially *shared* between the generative and representational models, rather than just being used in the encoder, while the KL divergence in our framework is taken before, not after, the mapping. Moreover, the underlying motivation, and type of mapping typically used, differs substantially: our mapping is used to introduce inductive biases, not purely to improve inference. Our mapping is also more general than a NF (e.g. it need not be invertible) and does not introduce additional constraints or computational issues.

## 6 SPECIFIC REALIZATIONS OF THE INTEL-VAE FRAMEWORK

We now present several novel example InteL-VAEs, introducing various inductive biases through different choices of $g_\psi$. We will start with artificial, but surprisingly challenging, examples where some precise topological properties of the target distributions are known, incorporating them directly through a fixed $g_\psi$. We will then move onto experiments where we impose a fixed clustering inductive bias when training on image data, allowing us to learn InteL-VAEs that account effectively for multi-modality in the data distribution. Finally, we consider the example of learning sparse representations of high–dimensional data. Here we will see that it is imperative to exploit the ability of InteL-VAEs to learn aspects of $g_\psi$ during training, providing a flexible inductive bias framework, rather than a pre-fixed mapping. By comparing InteL-VAEs with strong baselines, we show that InteL-VAEs are effective in introducing these desired inductive biases, and consequently both improve generation quality and learn better data representations for downstream tasks. One note of particular importance is that we find that InteL-VAEs provide state-of-the-art performance for learning sparse VAE representations. A further example of using InteL-VAEs to learn hierarchical representations is presented in Appendix B, while full details on the various examples are given in Appendix C.

### 6.1 MULTIPLE–CONNECTIVITY

Data is often most naturally described on non-Euclidean spaces such as circles, e.g. wind directions (Mardia & Jupp, 2000), and other multiply-connected shapes, e.g. holes in disease databases (Liu et al., 1997). For reasons previously explained in Sec. 2, standard VAEs cannot practically model such topologies, which prevents them from learning generative models which match even the simplest data distributions with non-trivial topological structures, as shown in Fig. 4b.

Luckily, by designing $g_\psi$ to map the Gaussian prior to a simple representative distribution in a topological class, we can easily equip InteL-VAEs with the knowledge to approximate any data distributions with similar topological properties. Specifically, by defining $g_\psi$ as the orthogonal projection to $\mathbb{S}^1$, $g_\psi(z) = z/(\|z\|_2 + \epsilon)$, we map the Gaussian prior approximately to a uniform distribution to $\mathbb{S}^1$, where $\epsilon$ is a small positive constant to ensure the continuity of $g_\psi$ near the origin. From Rows 1 and 2 of Fig. 4,

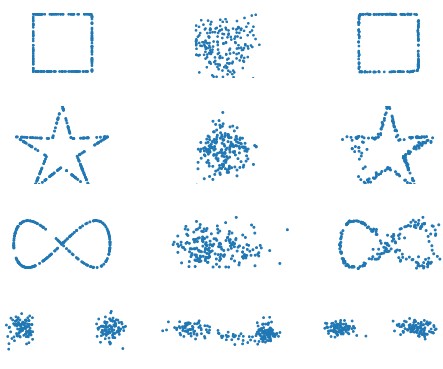

(a) Data     (b) VAE     (c) InteL-VAE

Figure 4: Training data and samples from learned generative models of vanilla-VAE and InteL-VAE for multiply-connected and clustered distributions. InteL-VAE uses [Rows 1,2] circular prior with one hole, [Row 3] multiply-connected prior with two holes, and [Row 4] clustered prior. Vamp-VAE behaves similarly to a vanilla VAE; its results are presented in Fig. 4.

we find that this inductive bias gives InteL-VAEs the ability to learn various distributions with a hole.

We can add further holes by simply 'gluing' point pairs. For example, for two holes we can use

$$g_2(y) = \text{Concat}\left(g_1(y)_{[:,1]},\ g_1(y)_{[:,2]}\sqrt{(4/3 - (1 - |g_1(y)_{[:,1]}|)^2)} - 1/\sqrt{3}\right), \qquad (5)$$

which first map $y$ to approximately $S^1$, and then glues $(0, 1)$ and $(0, -1)$ together to create new holes (see Fig. C.1 for an illustration). Furthermore, we can continue to glue points together to achieve a higher number of holes $h$, and thus more complex connectivity. Row 3 of Fig. 4 gives an example of learning an infinity sign by introducing a 'two-hole' inductive bias.

Compared with vanilla-VAE and Vamp-VAE, which try to find a convex hull for real data distributions, InteL-VAEs can deal with distributions with highly non-convex and very non-smooth supports (see Fig. 4 and Appendix C.1). We emphasize here that our inductive bias does not contain the information about the precise shape of the data, only the number of holes. We thus see that InteL-VAEs can provide substantial improvements in performance by incorporating only basic prior information about the topological properties of the data, which point out a way to approximate distributions on more complex structures, such as linear groups (Gupta & Mishra, 2018).

## 6.2 MULTI–MODALITY

Many real-world datasets exhibit multi-modality. For example, data with distinct classes are often naturally clustered into (nearly) disconnected components representing each class. However, vanilla VAEs generally fail to fit multi-modal data due to the topological issues explained in Sec. 2. Previous work (Johnson et al., 2017; Mathieu et al., 2019b) has thus proposed the use of a multi-modal prior, such as a mixture of Gaussian (MoG) distribution, so as to capture all components of the data. Nonetheless, VAEs with such priors often still struggle to model multi-modal data because of mismatch between $q_\phi(z)$ and $p(z)$ or training instability issues.

We tackle this problem by using a mapping $g_\psi$ which contains a clustering inductive bias. The high-level idea is to design a mapping $g_\psi$ with a localized high Lipschitz constant that 'splits' the continuous Gaussian distribution into $K$ disconnected parts and then pushes them away from each other. In particular, we split $\mathcal{Y}$ it into $K$ equally sized sectors using its first two dimensions (noting it is not needed to split on all dimensions to form clusters), as shown in Fig. 5. For any point $y$, we can easily get the center direction $\text{r}(y)$ of the sector that $y$ belongs to and the distance $\text{dis}(y)$ between $y$ and the sector boundary. Then we define $g_\psi(y)$ as:

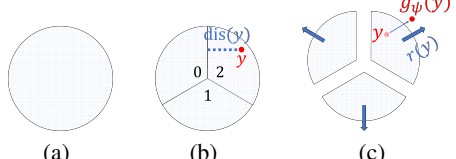

(a) (b) (c)

Figure 5: Illustration of clustered mapping where $K = 3$. The circle represents a density isoline of a Gaussian. Note that not all points in the sector are moved equally: points close to the boundaries between sectors are moved less, with points on the boundary themselves not moved at all.

$$g_\psi(y) = y + c_1\text{dis}(y)^{c_2}\text{r}(y), \qquad (6)$$

where $c_1$ and $c_2$ are empirical constants. We can see that although $g_\psi$ has very different function on different sectors, it is still continuous on the whole plane with $g_\psi(y) = y$ on sector boundaries, which is desirable for gradient-based training. See Appendix C.2 for more details.

To assess the performance of our approach, we first consider a simple 2-component MoG synthetic dataset in the last row of Fig. 4. We see that the vanilla VAE fails to learn a clustered distribution that fits the data, while the InteL-VAE sorts this issue and fits the data well.

To provide a more real-world example, we train an InteL-VAE and a variety of baselines on the **MNIST** dataset, comparing the generation quality of the learned models using the FID score (Heusel et al., 2017) in Table 1. We find that the GM-VAE (Dilok-thanakul et al., 2016) and MoG-VAE (VAE with a fixed MoG prior) achieve performance gains by using non-Gaussian priors. The Vamp-VAE (Tomczak & Welling, 2018) and a VAE with a Sylvester Normalizing Flow (Berg et al., 2018) encoder provide further gains by making the prior and encoder distributions more flexible respectively. However, the InteL-VAE comfortably outperforms all of them.

| Method | FID Score ($\downarrow$) |
| --- | --- |
| VAE | $42.0 \pm 1.1$ |
| GM-VAE | $41.0 \pm 4.7$ |
| MoG-VAE | $41.2 \pm 3.3$ |
| Vamp-VAE | $38.8 \pm 2.4$ |
| VAE with Sylvester NF | $35.0 \pm 0.9$ |
| InteL-VAE | $32.2 \pm 1.5$ |

Table 1: Generation quality on MNIST. Shown is mean FID score (lower better) $\pm$ standard deviation over 10 runs.

| Method | Data | VAE | GM-VAE | MoG-VAE | Vamp-VAE | Flow | InteL-VAE |
|---|---|---|---|---|---|---|---|
| **Uncertainty**(%) | $0.2 \pm 0.1$ | $2.5 \pm 0.4$ | $3.5 \pm 1.8$ | $4.5 \pm 0.8$ | $2.4 \pm 0.3$ | $16.2 \pm 2.1$ | $\mathbf{0.9 \pm 0.8}$ |
| **'1' proportion**(%) | $50.0 \pm 0.2$ | $48.8 \pm 0.2$ | $48.1 \pm 0.3$ | $47.7 \pm 0.4$ | $48.8 \pm 0.1$ | $42.5 \pm 1.0$ | $\mathbf{49.5 \pm 0.4}$ |

Table 2: Quantitative results on **MNIST-01**. **Uncertainty** is the proportion of images whose labels are 'indistinguishable' by the pre-trained classifier, defined as having prediction confidence $< 80\%$. **'1' proportion** is the proportion of images classified as '1'.

To gain insight into how InteL-VAEs achieve superior generation quality, we perform analysis on a simplified setting where we select only the '0' and '1' digits from the **MNIST** dataset to form a strongly clustered dataset, **MNIST-01**. We further decrease the latent dimension to 1 to make the problem more challenging. Fig. 6 shows that here the vanilla VAE generates

(a) VAE    (b) MoG-VAE    (c) InteL-VAE

Figure 6: Generated samples for **MNIST-01**.

some samples which look like interpolations between '0' and '1', meaning that it still tries to learn a connected distribution containing '0' and '1'. Further, the general generation quality is poor, with blurred images and a lack of diversity in generated samples (e.g. all the '1's have the same slant). Despite using a clustered prior, the MoG-VAE still produces unwanted interpolations between the classes. By contrast, InteL-VAE generates digits that are unambiguous and crisper.

To quantify these results, we further train a logistic classifier on **MNIST-01** and use it to classify images generated by each method. For each method, we calculate the proportion of samples produced by the generative model that are assigned to each class by this pre-trained classifier, as well as the proportion of samples for which the classifier is uncertain. From Table 2 we see that InteL-VAE significantly outperforms its competitors in the ability to generate balanced and unambiguous digits. To extend this example further, and show the ability of InteL-VAEs to learn aspects of $g_\psi$ during training, we further consider parameterizing and then learning the relative size of the clusters. Table 3 shows that this can be successfully learned by InteL-VAEs on **MNIST-01**.

| True Prop. | Learned Prop. |
|---|---|
| 0.5 | $0.47 \pm 0.01$ |
| 0.4 | $0.36 \pm 0.10$ |
| 0.25 | $0.25 \pm 0.08$ |
| 0.2 | $0.16 \pm 0.11$ |
| 0 | $0.02 \pm 0.01$ |

Table 3: Learned proportions of '0's on **MNIST-01** for different ground truths. Error bars are std. dev. from 10 runs.

### 6.3 SPARSITY

Sparse features are often well-suited to data efficiency on downstream tasks (Huang & Aviyente, 2006), in addition to being naturally easier to visualize and manipulate than dense features (Ng et al., 2011). However, existing VAE models for sparse representations trade off generation quality to achieve this sparsity (Mathieu et al., 2019b; Tonolini et al., 2020; Barello et al., 2018). Here, we show that InteL-VAEs can instead *simultaneously* increase feature sparsity and generation quality. Moreover, they are able to achieve state-of-the-art scores on sparsity metrics.

Compared with our previous examples, the $g_\psi$ here needs to be more flexible so that it can learn to map points in a data-specific way and induce sparsity without unduly harming reconstruction. To achieve this, we use the simple form for the mapping: $g_\psi(y) = y \odot \text{DS}_\psi(y)$, where $\odot$ is pointwise multiplication, and DS is a 'dimension selector' network that selects dimensions to deactivate given $y$. DS outputs values between $[0, 1]$ for each dimension, with 0 being fully deactivated and 1 fully activated; the more dimensions we deactivate, the sparser the representation. By learning DS during training, this setup allows us to learn a sparse representation in a data-driven manner. To control the degree of sparsity, we add a sparsity regularizer, $\mathcal{L}_{sp}$, to the ELBO with weighting parameter $\gamma$ (higher $\gamma$ corresponds to more sparsity). Namely, we optimize $\mathcal{L}_{\mathcal{Y}}(\theta, \phi, \psi) + \gamma \mathcal{L}_{sp}(\phi, \psi)$, where

$$\mathcal{L}_{sp}(\phi, \psi) := \mathbb{E}\left[ \frac{1}{M} \sum_{i=1}^{M} (H\left(DS(y_i)\right)) - H\left( \frac{1}{M} \sum_{i=1}^{M} DS(y_i) \right) \right], \qquad (7)$$

$H(v) = -\sum_i (v_i/\|v\|_1) \log (v_i/\|v\|_1)$ is the normalized entropy of an positive vector $v$, and the expectation is over drawing a minibatch of samples $x_1, \ldots, x_M$ and then sampling each corresponding $y_i \sim q_\phi(\cdot|x = x_i)$. $\mathcal{L}_{sp}$ encourages DS to deactivate more dimensions, while also encouraging diversity in which dimensions are activated for different data points, improving utilization of the latent space. Please see Appendix C.3 for more details and intuitions. Initial qualitative results are shown in Fig. 8, where we see that our InteL-VAE is able to learn sparse and intuitive representations.

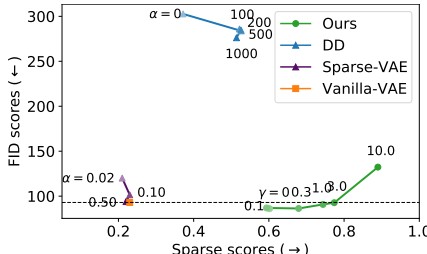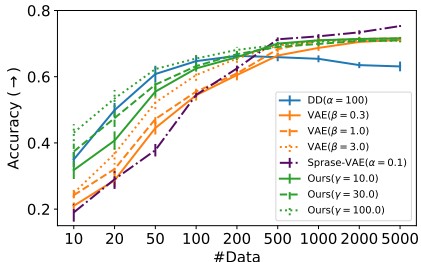

Figure 7: Results on **Fashion-MNIST**. The *left* figure shows FID and sparsity scores. Lower FID scores ($\downarrow$) represent better sample quality while higher sparse scores ($\rightarrow$) indicate sparser features. The *right* figure shows the performance of sparse features from InteL-VAE on downstream classification tasks. See Appendix C.3 for details and results for **MNIST**.

To quantitatively assess the ability of our approach to yield sparse representations and good quality generations, we compare against vanilla VAEs, the specially customized sparse-VAE of Tonolini et al. (2020), and the sparse version of Mathieu et al. (2019b) (DD) on **Fashion-MNIST** (Xiao et al., 2017) and **MNIST**. As shown in Fig. 7 (*left*), we find that InteL-VAEs increase sparsity of the representations—measured by the Hoyer metric (Hurley & Rickard, 2009)—while increasing generative sample quality at the same time. Indeed, the FID score obtained by InteL-VAE outperforms the vanilla VAE when $\gamma < 3.0$, while the sparsity score substantially increases with $\gamma$, reaching extremely high levels. By comparison, DD significantly degrades generation quality and only provides a more modest increase in sparsity, while its sparsity also drops if the regularization coefficient is set too high. The level of sparsity achieved by sparse-VAEs was substantially less than both DD and InteL-VAEs.

To further evaluate the quality of the learned features for downstream tasks, we trained a classifier to predict class labels from the latent representations. For this, we choose a random forest (Breiman, 2001) with maximum depth 4 as it is well-suited for sparse features. We vary the size of training data given to the classifier to measure the data efficiency of each model. Fig. 7 (*right*) shows that InteL-VAE typically outperforms other the models, especially in few-shot scenarios.

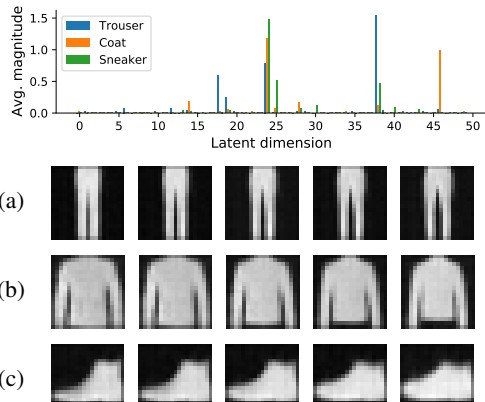

Figure 8: Qualitative evaluation of sparsity. [Top] Average magnitude of each latent dimension for three example classes in **Fashion-MNIST**; less than $10\%$ dimensions are activated for each class. [Bottom] Activated dimensions are different between classes: (a-c) show the results of separately manipulating an activated dimension for each class. (a) Trouser separation (Dim 18). (b) Coat length (Dim 46). (c) Shoe style (formal/sport, Dim 25).

Finally, to verify InteL-VAE's effectiveness on larger and higher-resolution datasets, we also make comparisons on **CelebA** (Liu et al., 2015). From Table 4, we can see that InteL-VAE increase sparse scores to 0.46 without sacrificing generation quality. By comparison, the maximal sparse score that sparse-VAE gets is 0.30, with unacceptable sample quality. Interestingly, InteL-VAEs with ly low regulation $\gamma$ achieved particularly good generative sample quality, outperforming even the Vamp-VAE and a VAE with a Sylvester NF encoder.

| Method | FID ($\downarrow$) | Sparsity ($\uparrow$) |
|---|---|---|
| VAE | 68.6±1.1 | 0.22±0.01 |
| Vamp-VAE | 67.5±1.1 | 0.22±0.01 |
| VAE with Sylvester NF | 66.3±0.4 | 0.22±0.01 |
| Sparse-VAE ($\alpha = 0.01$) | 328±10.1 | 0.25±0.01 |
| Sparse-VAE ($\alpha = 0.2$) | 337±8.1 | 0.28±0.01 |
| InteL-VAE ($\gamma = 30$) | 64.9±0.4 | 0.25±0.01 |
| InteL-VAE ($\gamma = 70$) | 68.0±0.6 | 0.46±0.02 |

Table 4: Generation results on CelebA.

**Conclusions** In this paper, we proposed InteL-VAEs, a general schema for incorporating inductive biases into VAEs. Experiments show that InteL-VAEs can both provide representations with desired properties and improve generation quality, outperforming a variety of baselines such as directly changing the prior. This is achieved while maintaining the simplicity and stability of standard VAEs.

## ETHICS STATEMENT

We do not believe that there are direct ethical concerns regarding our paper: the datasets we consider are all already well established and do not contain sensitive information, while the methods and ideas we introduce have no clear direct potential negative societal impacts of their own. From a bigger picture perspective, work like ours that looks to permit more effective incorporation of inductive biases into models can be thought of as allowing more direct human control on how models will behave after training. While this will typically be a force for good, for example by encouraging model interpretability and providing mechanisms to try and induce positive characteristics like fairness, in rare circumstances there may also be the potential for this to be used nefariously by deliberately encouraging undesirable behavior. However, we do not believe that our work is any more prone to such exploitation than existing methods or that the risk of it being used in such as a way is significant.

## REPRODUCIBILITY STATEMENT

Full experimental details are given in Appendix C, while anonymized source code for reproducing all our experiments directly is provided at https://github.com/djkdsjwkjerkjermf/InteL-VAE. Together these should make it straightforward for others to reproduce our empirical results. We have been careful to provide quantitative metrics of performance whenever possible, rather than just relying on qualitative or anecdotal evidence. Repeat runs and error bars are provided whenever this is feasible, with the level of variability always found to be sufficiently small to draw reliable and statistically sound conclusions. In fact, the training stability and consistent performance of our general approach under retraining provides a clear advantage in itself compared to many of the baseline methods. Full formal proof for our only theoretical result is given in Appendix A, while the assumptions it makes are clearly stated and easily verifiable.

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

## APPENDIX A    PROOFS

**Theorem 1.** *Let $p_\psi(z)$ and $q_{\phi,\psi}(z|x)$ represent the respective pushforward distributions of $\mathcal{N}(0, I)$ and $q_\phi(y|x)$ induced by the mapping $g_\psi : \mathcal{Y} \mapsto \mathcal{Z}$. The following holds for all measurable $g_\psi$:*

$$D_{KL}\left(q_{\phi,\psi}(z|x) \,\|\, p_\psi(z)\right) \leq D_{KL}\left(q_\phi(y|x) \,\|\, \mathcal{N}(y; 0, I)\right). \tag{3}$$

*If $g_\psi$ is also an invertible function then the above becomes an equality and $\mathcal{L}_\mathcal{Y}$ equals the standard ELBO on the space of $\mathcal{Z}$ as follows*

$$\mathcal{L}_\mathcal{Y}(x, \theta, \phi, \psi) = \mathbb{E}_{q_{\phi,\psi}(z|x)}[\log p_\theta(x|z)] - D_{KL}\left(q_{\phi,\psi}(z|x) \,\|\, p_\psi(z)\right). \tag{4}$$

*Proof.* We first prove the inequality from Eq. (3), then we show that Eq. (3) is actually an equality when $g_\psi$ is invertible, and finally we prove that the reconstruction term is unchanged by $g_\psi$.

Let us denote by $\mathcal{F}$ and $\mathcal{G}$ the sigma-algebras of respectively $\mathcal{Y}$ and $\mathcal{Z}$, and we have by construction a measurable map $g_\psi : (\mathcal{Y}, \mathcal{F}) \to (\mathcal{Z}, \mathcal{G})$. We can actually define the measurable space $(\mathcal{Z}, \mathcal{G})$ as the image of $(\mathcal{Y}, \mathcal{F})$ by $g_\psi$, then $g_\psi$ is automatically both surjective and measurable.[1] We also assume that there exists a measure on $\mathcal{Y}$, which we denote $\xi$, and denote with $\nu$ the corresponding pushforward measure by $g_\psi$ on $\mathcal{Z}$. We further have $\nu(A) = \xi(g_\psi^{-1}(A))$ for any $A \in \mathcal{G}$.[2]

We start by proving Eq. (3), where the Kullback-Leibler (KL) divergence between the two pushforward measures[3] $q_{\phi,\psi} \triangleq q_\phi \circ g_\psi^{-1}$ and $p_\psi \triangleq p \circ g_\psi^{-1}$ is upper bounded by $D_{KL}\left(q_\phi(y|x) \,\|\, p(y)\right)$, where here we have $p(y) = \mathcal{N}(y; 0, I)$ but we will use $p$ as a convenient shorthand. At a high-level, we essentially have that Eq. (3) follows directly the data processing inequality (Sason, 2019) with a deterministic kernel $z = g_\psi(y)$. Nonetheless, we develop in what follows a proof which additionally gives sufficient conditions for when this inequality becomes non-strict. We can assume that $D_{KL}\left(q_\phi(y|x) \,\|\, \mathcal{N}(y; 0, I)\right)$ is finite, as otherwise the result is trivially true, which in turn implies $q_\phi \ll p$.[4] For any $A \in \mathcal{G}$, we have that if $p_\psi(A) = p \circ g_\psi^{-1}(A) = p(g_\psi^{-1}(A)) = 0$ then this implies $q_\phi(g_\psi^{-1}(A)) = q_\phi \circ g_\psi^{-1}(A) = q_{\phi,\psi}(A) = 0$. As such, we have that $q_{\phi,\psi} \ll p_\psi$ and so the $D_{KL}\left(q_{\phi,\psi}(z|x) \,\|\, p_\psi(z)\right)$ is also defined.

Our next significant step is to show that

$$\mathbb{E}_{p(y)}\left[\frac{q_\phi}{p} \,\Big|\, \sigma(g_\psi)\right] = \frac{q_\phi \circ g_\psi^{-1}}{p \circ g_\psi^{-1}} \circ g_\psi, \tag{A.1}$$

where $\sigma(g_\psi)$ denotes the sigma-algebra generated by the function $g_\psi$. To do this, let $h : (\mathcal{Z}, \mathcal{G}) \to (\mathbb{R}_+, \mathcal{B}(\mathbb{R}_+))$ be a measurable function s.t. $\mathbb{E}_{p(y)}\left[\frac{q_\phi}{p} \,\Big|\, \sigma(g_\psi)\right] = h \circ g_\psi$. To show this, we will demonstrate that they lead to equivalent measures when integrated over any arbitrary set $A \in \mathcal{G}$:

$$\int_\mathcal{Z} \mathbb{1}_A \frac{q_\phi \circ g_\psi^{-1}}{p \circ g_\psi^{-1}} \, p \circ g_\psi^{-1} \, d\nu = \int_\mathcal{Z} \mathbb{1}_A \, q_\phi \circ g_\psi^{-1} \, d\nu = \int_\mathcal{Z} \mathbb{1}_A \, d(q_\phi \circ g_\psi^{-1})$$

$$\stackrel{(a)}{=} \int_\mathcal{Y} (\mathbb{1}_A \circ g_\psi) \, dq_\phi = \int_\mathcal{Y} (\mathbb{1}_A \circ g_\psi) \, q_\phi \, d\xi$$

$$\stackrel{(b)}{=} \int_\mathcal{Y} (\mathbb{1}_A \circ g_\psi) \, \frac{q_\phi}{p} \, p \, d\xi$$

$$\stackrel{(c)}{=} \int_\mathcal{Y} (\mathbb{1}_A \circ g_\psi) \, \mathbb{E}_{p(y)}\left[\frac{q_\phi}{p} \,\Big|\, \sigma(g_\psi)\right] \, p \, d\xi$$

$$\stackrel{(d)}{=} \int_\mathcal{Y} (\mathbb{1}_A \circ g_\psi) \, (h \circ g_\psi) \, p \, d\xi = \int_\mathcal{Y} (\mathbb{1}_A \circ g_\psi) \, (h \circ g_\psi) \, dp$$

$$\stackrel{(e)}{=} \int_\mathcal{Z} \mathbb{1}_A \, h \, d(p \circ g_\psi^{-1}) = \int_\mathcal{Z} \mathbb{1}_A \, h \, (p \circ g_\psi^{-1}) \, d\nu,$$

---

[1] We recall that $g_\psi$ is said to be measurable if and only if for any $A \in \mathcal{G}$, $g_\psi^{-1}(A) \in \mathcal{F}$.

[2] The notation $g_\psi^{-1}(A)$ does not imply that $g_\psi$ is invertible, but denotes the preimage of $A$ which is defined as $g_\psi^{-1}(A) = \{y \in \mathcal{Y} \mid g_\psi(y) \in A\}$.

[3] We denote the pushforward of a probability measure $\chi$ along a map $g$ by $\chi \circ g^{-1}$.

[4] We denote the absolute continuity of measures with $\ll$, where $\mu$ is said to be absolutely continuous w.r.t. $\nu$, i.e. $\mu \ll \nu$, if for any measurable set $A$, $\nu(A) = 0$ implies $\mu(A) = 0$.

where we have leveraged the definition of pushforward measures in (a & e); the absolute continuity of $q_\phi$ w.r.t. $p$ in (b); the conditional expectation definition in (c); and the definition of $h$ in (d). By equating terms, we have that $q_\phi \circ g_\psi^{-1} / p \circ g_\psi^{-1} = h$, almost-surely with respect to $q_\phi \circ g_\psi^{-1}$ and thus that Eq. (A.1) is verified.

Let us define $f : x \mapsto x \log(x)$, which is strictly convex on $[0, \infty)$ (as it can be prolonged with $f(0) = 0$). We have the following

$$
\begin{aligned}
D_{\mathrm{KL}}\left(q_{\phi,\psi}(z|x) \| p_\psi(z)\right) &\stackrel{(a)}{=} \int_{\mathcal{Z}} \log\left(\frac{q_{\phi,\psi}}{p_\psi}\right) q_{\phi,\psi}\, d\nu \\
&\stackrel{(b)}{=} \int_{\mathcal{Z}} \log\left(\frac{q_{\phi,\psi}}{p_\psi}\right) \frac{q_{\phi,\psi}}{p_\psi}\, p_\psi\, d\nu \\
&\stackrel{(c)}{=} \int_{\mathcal{Z}} f\left(\frac{q_{\phi,\psi}}{p_\psi}\right) p_\psi\, d\nu = \int_{\mathcal{Z}} f\left(\frac{q_{\phi,\psi}}{p_\psi}\right) d(p \circ g_\psi^{-1}) \\
&\stackrel{(d)}{=} \int_{\mathcal{Y}} f\left(\frac{q_{\phi,\psi}}{p_\psi} \circ g_\psi\right) dp = \int_{\mathcal{Y}} f\left(\frac{q_\phi \circ g_\psi^{-1}}{p \circ g_\psi^{-1}} \circ g_\psi\right) p\, d\xi \\
&\stackrel{(e)}{=} \int_{\mathcal{Y}} f\left(\mathbb{E}_{p(y)}\left[\frac{q_\phi}{p} \,\middle|\, \sigma(g_\psi)\right]\right) p\, d\xi \\
&\stackrel{(f)}{\leq} \int_{\mathcal{Y}} \mathbb{E}_{p(y)}\left[f\left(\frac{q_\phi}{p}\right) \,\middle|\, \sigma(g_\psi)\right] p\, d\xi \\
&\stackrel{(g)}{=} \int_{\mathcal{Y}} f\left(\frac{q_\phi}{p}\right) p\, d\xi \\
&\stackrel{(h)}{=} \int_{\mathcal{Y}} \log\left(\frac{q_\phi}{p}\right) \frac{q_\phi}{p}\, p\, d\xi \\
&\stackrel{(i)}{=} \mathbb{E}_{q_\phi(y|x)}\left[\log\left(\frac{q_\phi(y|x)}{p(y)}\right)\right] \\
&\stackrel{(j)}{=} D_{\mathrm{KL}}\left(q_\phi(y|x) \| p(y)\right),
\end{aligned}
$$

where we leveraged the definition of the KL divergence in (a & j); the absolute continuity of $q_\phi$ w.r.t. $p$ in (b & i); the definition of $f$ in (c & h); the definition of the pushforward measure in (d); Eq. (A.1) in (e); the conditional Jensen inequality in (f) and the law of total expectation in (g). Note that this proof not only holds for the KL divergence, but for any f-divergences as they are defined as in (b) with $f$ convex.

To prove Eq. (4), we now need to show that line (f) above becomes an equality when $g_\psi$ is invertible. As $f$ is strictly convex, this happens if and only if $\frac{q_\phi}{p} = \mathbb{E}_{p(y)}\left[\frac{q_\phi}{p} \,\middle|\, \sigma(g_\psi)\right]$. A sufficient condition for this to be true is for $\frac{q_\phi}{p}$ to be measurable w.r.t. $\sigma(g_\psi)$ which is satisfied when $g_\psi : \mathcal{Y} \mapsto \mathcal{Z}$ is invertible as $\sigma(g_\psi) \supseteq \mathcal{F}$, as required. We have thus shown that the KL divergences are equal when using an invertible $g_\psi$.

For the reconstruction term, we instead have

$$
\begin{aligned}
\mathbb{E}_{q_\phi(y|x)}[\log p_\theta(x|g_\psi(y))] &= \int_{\mathcal{Y}} \log p_\theta(x|g_\psi(y)) q_\phi(y|x) d\xi \\
&= \int_{\mathcal{Z}} \log p_\theta(x|z) q_{\phi,\psi}(z|x) d\nu \\
&= \mathbb{E}_{q_{\phi,\psi}(z|x)}[\log p_\theta(x|z)].
\end{aligned}
$$

Eq. (4) now follows from the fact that both the reconstruction and KL terms are equal.

$\square$

## APPENDIX B  HIERARCHICAL REPRESENTATIONS

The isotropic Gaussian prior in standard VAEs assumes that representations are independent across dimensions (Kumar et al., 2018). However, this assumption is often unrealistic (Belghazi et al., 2018; Mathieu et al., 2019b). For example, in Fashion-MNIST, high-level features such as object category, may affect low-level features such as shape or height. Separately extracting such global and local information can be beneficial for visualization and data manipulation (Zhao et al., 2017). To try and capture this, we introduce an inductive bias that is tailored to model and learn hierarchical features. We note here that our aim is not to try and provide a state-of-the-art hierarchical VAE approach, as a wide variety of highly–customized and powerful approaches are already well–established, but to show how easily the InteL-VAE framework can be used to induce hierarchical representations in a simple, lightweight, manner.

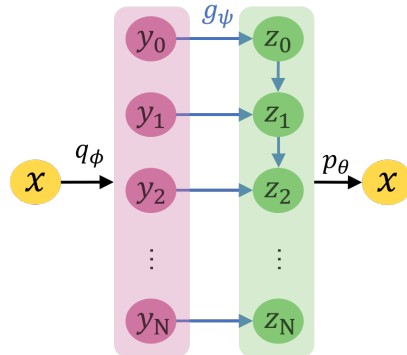

Figure B.1: Graphical model for hierarchical InteL-VAE

**Mapping design**    Following existing ideas from hierarchical VAEs (Sønderby et al., 2016; Zhao et al., 2017), we propose a hierarchical mapping $g_\psi$. As shown in Fig. B.1, the intermediary Gaussian variable $y$ is first split into a set of $N$ layers $[y_0, y_1, ..., y_N]$. The mapping $z = g_\psi(y)$ is then recursively defined as $z_i = \text{NN}_i(z_{i-1}, y_i)$, where $\text{NN}_i$ is a neural network combining information from higher-level feature $z_{i-1}$ and new information from $y_i$. As a result, we get a hierarchical encoding $z = [z_0, z_1, ..., z_N]$, where high-level features influence low-level ones but not vice-versa. This $g_\psi$ thus endows InteL-VAEs with hierarchical representations.

**Experiments**    While conventional hierarchical VAEs, e.g. (Sønderby et al., 2016; Zhao et al., 2017; Vahdat & Kautz, 2020), use hierarchies to try and improve generation quality, our usage is explicitly from the representation perspective, with our experiments set up accordingly. Fig. B.2 shows some hierarchical features learned by InteL-VAE on **Fashion-MNIST**. We observe that high-level information such as categories have indeed been learned in the top-level features, while low-level features control more detailed aspects.

To provide more quantitative investigation, we also consider the **CelebA** dataset (Liu et al., 2015) and investigate performance on downstream tasks, comparing to vanilla-VAEs with different latent dimensions. For this, we train a linear classifier to predict all 40 binary labels from the learned features for each method. In order to eliminate the effect of latent dimensions, we compare InteL-VAE (with fixed latent dimension 128) and vanilla VAE with different latent dimensions $(1, 2, 4, 8, 16, 32, 64, 128)$. We show experiment results on some labels as well as the average accuracy on all labels in Table B.1 and Fig. B.3. We first find that the optimal latent dimension increases

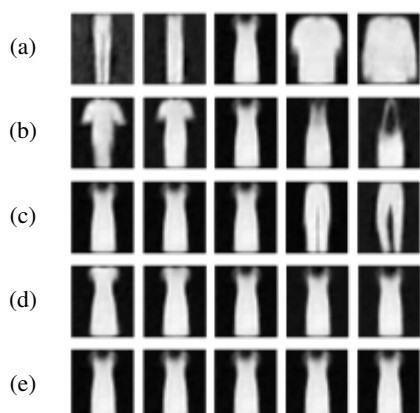

Figure B.2: Manipulating representations of a hierarchical InteL-VAE. The features are split into 5 levels, with each of (a) [highest] to (e) [lowest] corresponding to an example feature from each. We see that high-level features control more complex properties, such as class label or topological structure, while low-level features control simpler details, (e.g. (d) controls collar shape).

with the number of data points for the vanilla-VAEs, but is always worse than the InteL-VAE. Notably, the accuracy with InteL-VAE is quite robust, even as the number of data points gets dramatically low, indicating high data efficiency. To the best of our knowledge, this is the first result showing that a hierarchical inductive bias in VAE is beneficial to feature quality.

**Related work**    Hierarchical VAEs (Vahdat & Kautz, 2020; Ranganath et al., 2016; Sønderby et al., 2016; Klushyn et al.; Zhao et al., 2017) seek to improve the fit and generation quality of VAEs by recursively correcting the generative distributions. However, they require careful design of neural

| Model | Latent dim | Data size | | | | | |
|---|---|---|---|---|---|---|---|
| | | 50 | 100 | 500 | 1000 | 5000 | 10000 |
| VAE | 8 | 0.791 | 0.799 | 0.814 | 0.815 | 0.819 | 0.819 |
| | 16 | 0.788 | 0.801 | 0.820 | 0.824 | 0.829 | 0.831 |
| | 32 | 0.769 | 0.795 | 0.825 | 0.832 | 0.842 | 0.846 |
| | 64 | 0.767 | 0.794 | 0.826 | 0.832 | 0.849 | 0.855 |
| | 128 | 0.722 | 0.765 | 0.817 | 0.825 | 0.830 | 0.852 |
| InteL-VAE | 64 | **0.817** | **0.824** | **0.841** | **0.846** | **0.854** | **0.857** |

Table B.1: Average accuracy in predicting all 40 binary labels of **CelebA**. Overall best accuracy is shown in bold and best results of vanilla-VAEs are underlined for comparison. Each experiment is repeated 10 times and differences are significant at the $5\%$ level for data size $\leq 1000$.

layers, and the hierarchical KL divergence makes training deep hierarchical VAEs unstable (Vahdat & Kautz, 2020). In comparison, InteL-VAE with hierarchical mappings is extremely easy to implement without causing any computational instabilities, while its aims also differ noticeably: our approach successfully learns hierarchical *representations*—something that is rarely mentioned in prior works.

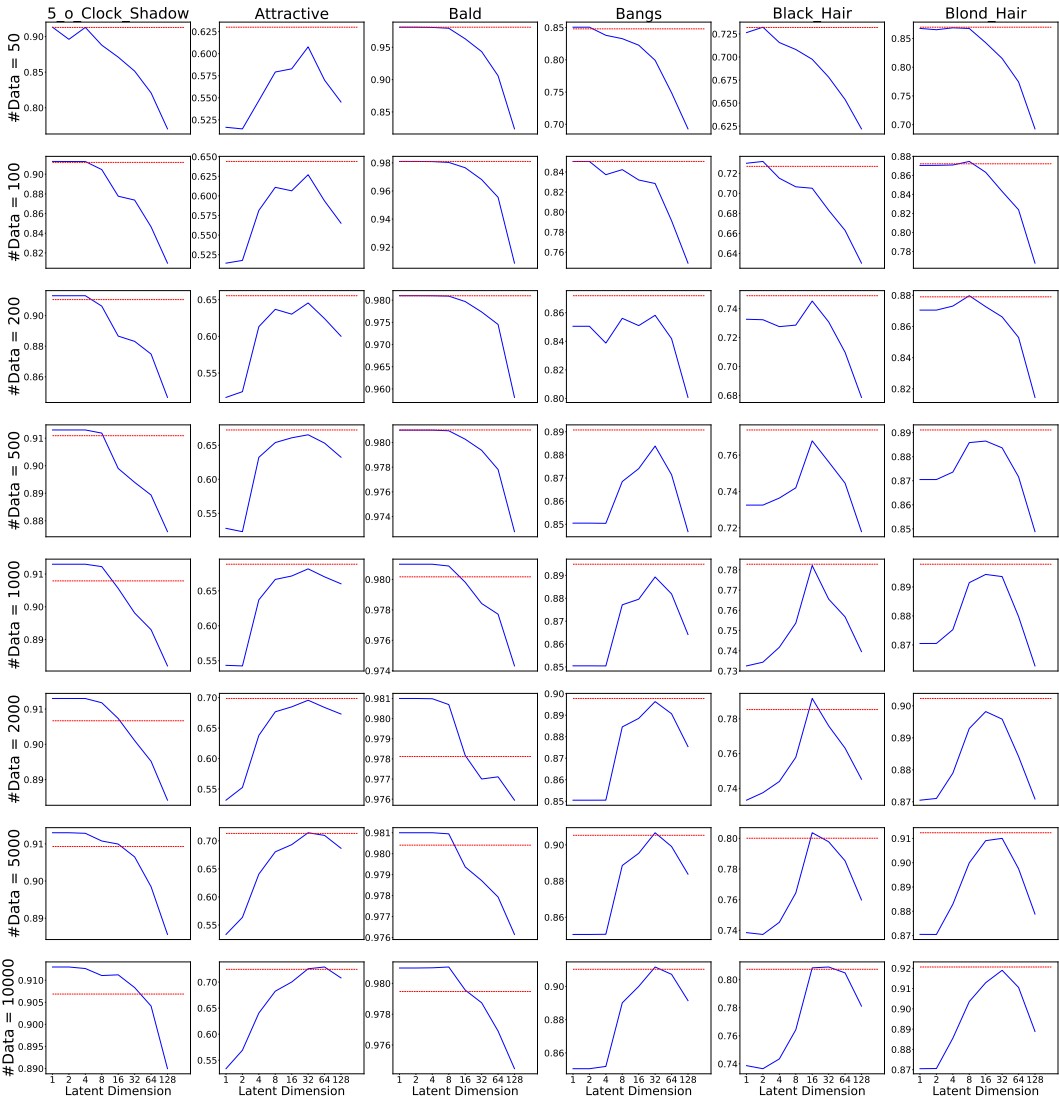

Figure B.3: InteL-VAE's performance of attribute prediction on **CelebA** dataset. Each column shows results on the same feature with different data sizes and each column shows results on different features. In each graph, test accuracy of vanilla-VAE with different latent dimensions are shown in blue line. And results of InteL-VAE with hierarchical prior are shown in red. We find that our method (red line) achieves comparable or even better results compared with vanilla-VAE with all latent dimensions.

## APPENDIX C   FULL METHOD AND EXPERIMENT DETAILS

In this section, we first provide complete details of the mapping designs used for our different InteL-VAE realizations along with some additional experiments. We then provide other general information about datasets, network structures, and experiment settings to facilitate results reproduction.

### C.1   MULTIPLE-CONNECTIVITY

**Mapping design**   Full details for this mapping were given in the main paper. Fig. C.1 provides a further illustration of the gluing process. Additional resulting including the Vamp-VAE are given in Fig. 4.

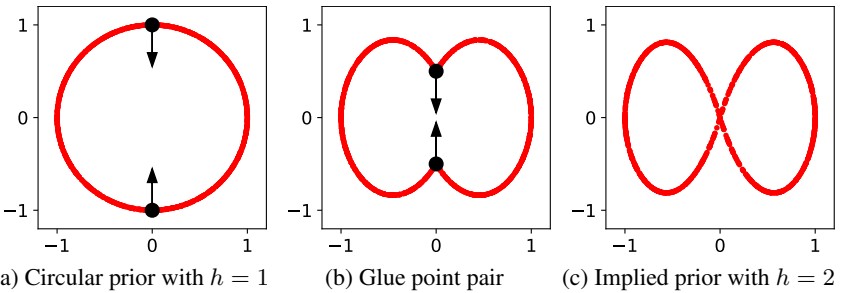

(a) Circular prior with $h = 1$    (b) Glue point pair    (c) Implied prior with $h = 2$

Figure C.1: An illustration of the glue function in multiply-connected mappings.

### C.2   MULTI-MODALITY

**Mapping design**   In Sec. 6.2, we see the general idea of designing clustered mappings. In this part, we delve into the details of mapping design as well as extending it to 1 dimensional and high-dimensional cases. For simplicity's sake let us temporarily assume that the dimension of $\mathcal{Y}$ is 2. Our approach is based on splitting the original space into $K$ equally sized sectors, where $K$ is the number of clusters we wish to create, as shown in Fig. 5b. For any point $y$, we can get its component (sector) index $\text{ci}(y)$ as well as its distance from the sector boundary $\text{dis}(y)$. By further defining the radius direction for the $k$-th sector (cf Fig. 5c) as

$$\Delta(k) = \left( \cos\left( \frac{2\pi}{K}\left( k + \frac{1}{2} \right) \right), \sin\left( \frac{2\pi}{K}\left( k + \frac{1}{2} \right) \right) \right) \quad \forall k \in \{1, \ldots, K\},$$

we can in turn define $g(y)$ as:

$$\text{r}(y) = \Delta(\text{ci}(y)), \tag{C.1}$$
$$g(y) = y + c_1\text{dis}(y)^{c_2}\text{r}(y), \tag{C.2}$$

where $c_1$ and $c_2$ are constants, which are set to 5 and 0.2 in our experiments. we make sure $g$ still continuous by keeping $g(y) = y$ on boundaries.

When dimension of $\mathcal{Y}$ is greater than 2, we have more diverse choice for $g$. When $K$ is decomposable, i.e., $K = \prod_i K_i$, we can separately cut the plane expanded by $\mathcal{Y}_{2i}$ and $\mathcal{Y}_{2i+1}$ into $K_i$ sectors by the Eq. (C.1). As a result, $\mathcal{Y}$ is split into $K = \prod_i k_i$ clusters. When $K = 2$, we find that $g$ only changes the 1-st dimension of $\mathcal{Y}$, so it can be applied to cases where latent dimension is 1.

**Learnable proportions**   We can also make the mapping more flexible by learning rather than assigning the cluster proportions. To do so, we keep a learnable value $u_i$ for each cluster and set the angle of the $i$-th sector as $2\pi\text{Softmax}(u)_i$. Things are simpler for the 1-dimensional case where we can uniformly translate $y$ by a learnable bias $b$ before splitting the space from the origin.

### C.3   SPARSITY

**Relationship to soft attention**   We note that our setup for the sparsity mapping shares some similarities with a soft attention layer (Bahdanau et al., 2014). However, there are also some

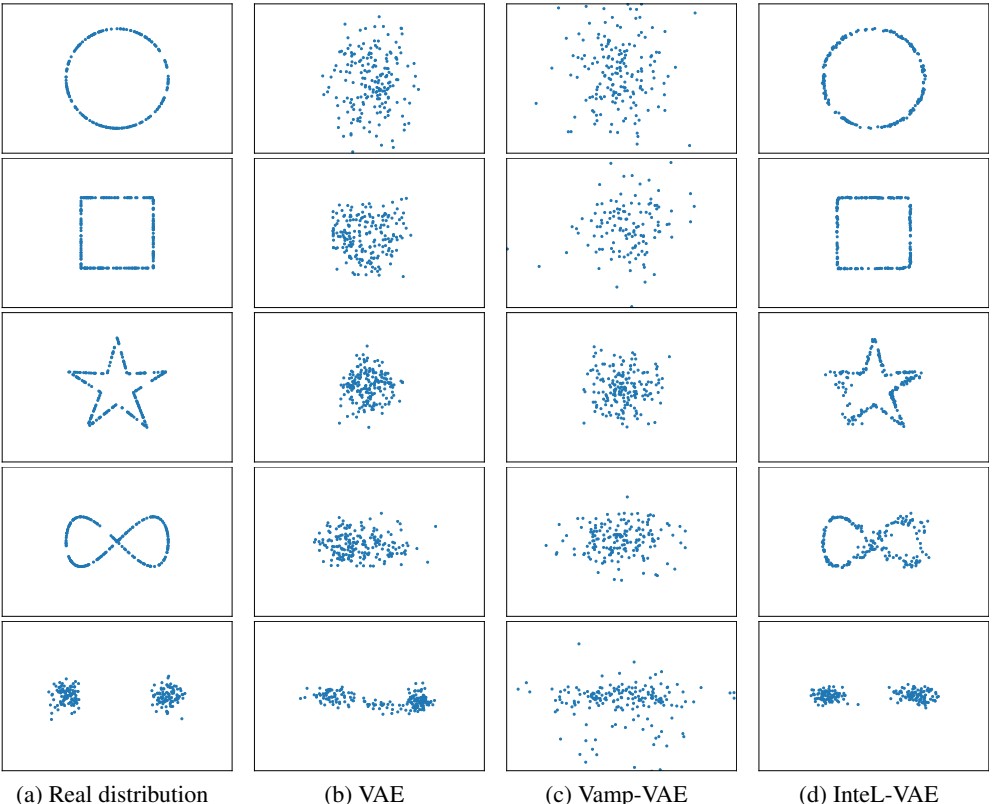

|                        |         |              |               |
|:----------------------:|:-------:|:------------:|:-------------:|
| (a) Real distribution  | (b) VAE | (c) Vamp-VAE | (d) InteL-VAE |

Figure C.2: Extension of Fig. 4 showing Vamp-VAE baseline and additional circular target distribution (top row, uses the same single hole $g_\psi$ as the second and third rows).

important points of difference. Firstly, soft attention aims to find the weights to blend features from different time steps (for sequential data) or different positions (for image data). In contrast, the dimension selector (DS) selects which dimensions to activate or deactivate for the same latent vector. Secondly, the weights of features are usually calculated by inner products of features for soft attention, while DS relies on a network to directly output the logits.

**Sparsity regularizer**    Our sparsity regularizer term, $\mathcal{L}_{sp}$, is used to encourage our dimensionality selector network (DS) to produce sparse mappings. It is defined using a mini-batch of samples $\{y_i\}_{i=1}^M$ drawn during training as per (7). During training, the first term of $\mathcal{L}_{sp}$ decreases the number of activated dimensions for each sample, while the second term prevents the samples from all using the same set of activated dimensions, which would cause the model to degenerate to a vanilla VAE with a lower latent dimensionality.

We note that $\mathcal{L}_{sp}$ alone is not expected to induce sparsity without also using the carefully constructed $g_\psi$ of the suggested InteL-VAE. We confirm this empirically by performing an ablation study on **MNIST** where we apply this regularization directly to a vanilla VAE. We find that even when using very large values of $\gamma > 30.0$ we can only slightly increase the sparsity score ($0.230 \rightarrow 0.235$). Moreover, unlikely for the InteL-VAE, this substantially deteriorates generation quality, with the FID score raising to more than $80.0$ at the same time.

**Sparse metric**    We use the Hoyer extrinsic metric (Hurley & Rickard, 2009) to measure the sparsity of representations. For a representation $z \in \mathbb{R}^D$,

$$\text{Hoyer}(z) = \frac{\sqrt{D} - ||\hat{z}||_1 / ||\hat{z}||_2}{\sqrt{D} - 1}. \tag{C.3}$$

Here, following Mathieu et al. (2019b), we crucially first normalized each dimension $d$ of $z$ to have standard deviation 1, $\hat{z}_d = z_d / \sigma_d$, to ensure that we only measure sparsity that varies between data

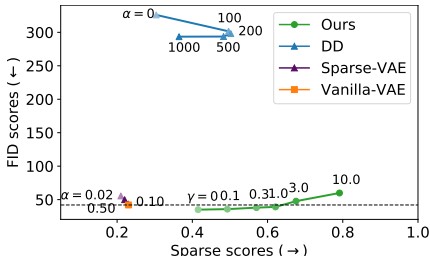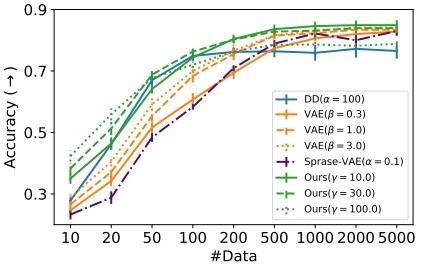

Figure C.3: Results on **MNIST**. The *left* figure shows FID and sparsity scores. Lower FID scores ($\downarrow$) represent better sample quality while higher sparse scores ($\rightarrow$) indicate sparser features. The *right* figure shows the performance of sparse features from InteL-VAE on downstream classification tasks. See Sec. 6.3 for details and results for **MNIST**.

| Parameters | Synthetic | MNIST | Fashion-MNIST | MNIST-01 | CelebA |
|---|---|---|---|---|---|
| Dataset sizes | Unlimited | 55k/5k/10k | 55k/5k/10k | 10k/1k/2k | 163k/20k/20k |
| Input space | $\mathbf{R}^2$ | Binary 28x28 | Binary 28x28 | Binary 28x28 | RGB 64x64x3 |
| Encoder net | MLP | CNN | CNN | CNN | CNN |
| Decoder net | MLP | CNN | CNN | CNN | CNN |
| Latent dimension | 2-10 | 50 | 50 | 1-10 | 1-128 |
| Batch size | 10-500 | 100 | 100 | 100 | 100 |
| Optimizer | Adam | Adam | Adam | Adam | Adam |
| Learning rate | 1e-3 | 1e-3 | 1e-3 | 1e-3 | 1e-3 |

Table C.1: Hyperparameters used for different experiments.

points (as is desired), rather than any tendency to uniformly 'switch off' certain latent dimensions (which is tangential to our aims). In other words, this normalization is necessary to avoid giving high scores to representations whose length scales vary between dimensions, but which are not really sparse.

By averaging Hoyer($z$) over all representations, we can get the sparse score of a method. For the sparsest case, where each representation has a single activated dimension, the sparse score is $1$. And when the representations get denser, $||\hat{z}||_2$ get smaller compared with $||\hat{z}||_1$, leading to smaller sparse scores.

**Reproduction of Sparse-VAE** We tried two different code bases for Sparse-VAE (Tonolini et al., 2020). The official code base[5] gives higher sparse scores for MNIST and FashionMNIST (though still lower than InteL-VAE), but is very unstable during training, with runs regularly failing after diverging and producing NaNs. This issue gets even more severe on CelebA which occurs after only a few training steps, undermining our ability to train anything meaningful at all. To account for this, we switched to the codebase[6] from De la Fuente & Aduviri (2019) that looked to replicate the results of the original paper. We report the results from this code base because it solves the instability issue and achieves reasonable results on CelebA. Interestingly, though its generation quality is good on MNIST and Fashion-MNIST, it fails to achieve a sparse score significantly higher than vanilla-VAE. As the original paper does not provide any quantitative evaluation of the achieved sparsity, it is difficult to know if this behavior is expected. We note though that the qualitative results shown in the paper appear to be substantially less sparse than those we show for the InteL-VAE, cf their Figure 5 compared to the top row of our Fig. 8. In particular, their representation seems to mostly 'switch off' some latents entirely, rather than having diversity between datapoints that is needed to score well under the Hoyer metric.

---

[5] https://github.com/ftonolini45/Variational_Sparse_Coding
[6] https://github.com/Alfo5123/Variational-Sparse-Coding

| Encoder | Decoder |
|---|---|
| Input 64 x 64 x 3 | Input $dim$ |
| 4x4 conv. 64 stride 2 & BN & LReLU | Dense (8x8x256) & BN & ReLU |
| 4x4 conv. 128 stride 2 & BN & LReLU | 4x4 upconv. 256 stride 2 & BN & ReLU |
| 4x4 conv. 256 stride 2 & BN & LReLU | 4x4 upconv. 128 stride 2 & BN & ReLU |
| Dense ($dim$) | 4x4 upconv. 3 stride 2 |

Table C.2: Encoder and Decoder structures for CelebA, where $dim$ is the latent dimension.

## C.4 ADDITIONAL EXPERIMENT DETAILS

**Datasets** Both synthetic and real datasets are used in this paper. All synthetic datasets (sphere, square, star, and mixture of Gaussian) are generated by generators provided in our codes. For real datasets, We load MNIST, Fashion-MNIST, and CelebA directly from Tensorflow (Abadi et al., 2015), and we resize images from CelebA to 64x64 following Hou et al. (2017). For experiments with a specified number of training samples, we randomly select a subset of the training data. We use the same random seed for each model in the same experiment and different random seeds when repeating experiments.

**Model structure** For low-dimensional data, the encoder and decoder are both simple multilayer perceptrons with 3 hidden layers (10-10-10) and ReLU (Glorot et al., 2011) activation. For MNIST and Fashion-MNIST, we use the same encoder and decoder as Mathieu et al. (2019b). For CelebA, the structure of convolutional networks are shown in Table C.2.

**Experiment settings** Other hyperparameters are shown in Table C.1. All experiments are run on a GTX-1080-Ti GPU.

