# OpenReview forum: "On Incorporating Inductive Biases into VAEs"
_ICLR.cc/2022/Conference — ICLR 2022 Poster_

### Official Review · Reviewer_pbbP · 2021-10-25

**Correctness:** 4
**Technical Novelty And Significance:** 3
**Empirical Novelty And Significance:** 3
**Recommendation:** 6
**Confidence:** 5

**Main Review:**

Strengths:
- Incorporating human knowledge into deep learning architectures is becoming a very relevant research area as the ML community is integrating with science and engineering. The paper offers a relevant contribution to this booming field of research.
- The paper is very well written and it contains sound and convincing arguments in favor of using user designed inductive biases and against the use of the VAE prior in order to induce those constraints.
- While the general approach is not very novel, the specific layers used to induce topological constraints are innovative, clever and useful. In my opinion the main contribution comes from these layers rather than the general technique.

Weaknesses:
- From a technical point of view, the main contribution is hardly novel as it simply entails the use of a user designed fist encoder layer. However, the approach is interesting from a conceptual and  interpretability standpoint.
- The sparsity application is hardly novel as it is nothing more than a simple soft attention layer. Soft attention is already widely used in the generative modeling literature.

Suggestions:
- I would recoment the authors to de-emphasize the general approach and to leave more space for the details of the topological layers used as in my opinion they are the main innovation of this work.

**Summary Of The Paper:**

This work introduces a new method to embed used-designed inductive-biases into VAE architectures. the basic idea is straightforward: a fixed user-defined deterministic transformation is used to map the latent z into a "structured latent" y. The transformation is used to induces topological and/or sparsity constraints. Technically, this is identical to using the transformation as the first layer of the decoder architecture. Besides of the general idea, the authors introduce a variety of simple but clever transformations used to enforce latent constraints. The approach is shown to offer an interpretable latent space and attractive performance in a large class of density estimation tasks.

**Summary Of The Review:**

The paper is well written and convincing. The new framework is not novel from a technical point of view but the details of the layers introduced to model complex topologies are. There is a major need of methods for embedding human knowledge into deep learning architectures and this paper is on a good direction.

---

> ### Author Response · Authors · 2021-11-15
> **Reply to Reviewer pbbP**
>
> We thank you for your helpful and insightful review and praise of our work.  We have made general updates of the submission in line with your suggestions and respond to some of the specific points raised below.
>
> &nbsp;
>
> >*I would recoment the authors to de-emphasize the general approach and to leave more space for the details of the topological layers.*
>
> We thank you for this useful advice! While we still believe the general approach is a very important component of our work as we discuss below, we wholeheartedly agree that the original submission of the paper did not sufficiently cover, and emphasize the novelty of, the particular definitions of $g_{\psi}$. As discussed in our general response, we have therefore made notable changes to the submission to provide more comprehensive coverage of this important aspect.  In particular, we now cover all the key details of the specific mappings in the main paper.
>
> &nbsp;
>
> >*From a technical point of view, the main contribution is hardly novel as it simply entails the use of a user designed first [decoder] layer. However, the approach is interesting from a conceptual and interpretability standpoint.*
>
> While we agree that algorithmically our approach is very simple and that many of our contributions are conceptual, we believe this is actually a strength of our work: we do not believe that our high-level approach was an obvious solution to the considered problem given previous work, and thus has significant technical novelty.
>
> Firstly, we note that previous papers that have suggested adding comparable inductive biases into VAEs have almost universally done so through direct changes to the prior (see e.g. second paragraph of the introduction); we believe none suggest something comparable to our own framework.  If the approach we suggest was the obvious solution to instilling inductive biases into VAE a priori, it stands to reason that these papers would have taken our suggested approach instead of an approach we show is significantly inferior. It is certainly the case that we ourselves took a long time to settle on the final suggested approach when trying to tackle this inductive bias problem.
>
> Second, as you allude to yourself along with other reviewers, the type of customized decoder layers we are suggesting are substantially different to those that have been used before, with the conceptual contributions of the work a necessary stepping stone to their introduction.
>
> Thirdly, even when taking the view of $g_{\psi}$ as the first layer of the decoder (cf our general response), using part of the decoder to construct *representations* with the desired characteristics / inductive biases is itself a novel high-level idea that is highly distinct to the general notion of incorporating prior information into the architecture design.
>
> &nbsp;
>
> >*The sparsity application is hardly novel as it is nothing more than a simple soft attention layer.*
>
> While we agree that the dimension selector (DS) in the sparse mapping shares some design features with soft attention, we note there are important points of difference between them. Firstly, soft attention aims to find the weights to blend features from different time steps (for sequential data) or different positions (for image data). In contrast, the DS selects which dimensions to activate or deactivate for the same latent vector. They thus vary quite noticeably in both motivation and in how they are applied. Secondly, the weights of features are usually calculated by inner products of features for soft attention, while DS relies on a network to directly output the logits. Finally, we believe the sparse regularizer *$L_{sp}$* to be novel, which more directly promotes sparsity without sacrificing diversity.
>
> We have added a paragraph on this relationship to Appendix C.3, along with a citation to the self attention paper.

---

> > ### Comment · Reviewer_pbbP · 2021-11-21
> > **Acknowledgment**
> >
> > I wish to thank the authors for the detailed response and the updates in the manuscript. I do think that the paper has improved and I would be happy to see it accepted.
> > However, I do stand by all points I raised in my review and I will therefore not change the original score.

---

> > > ### Author Response · Authors · 2021-11-24
> > > **Re: Acknowledgment**
> > >
> > > We are delighted that you like our work and find the paper improved. Thank you very much for your time reading the rebuttal and the updated manuscript.

---

### Official Review · Reviewer_NaAZ · 2021-10-26

**Correctness:** 2
**Technical Novelty And Significance:** 2
**Empirical Novelty And Significance:** 2
**Recommendation:** 6
**Confidence:** 4

**Main Review:**

Strengths:
- The idea of modifying decoders is interesting.
- The presented mappings seem to be indeed helpful in most of the cases presented in the experimental section.

Weaknesses:
- I do not agree with the statement that the prior in the VAE framework is about regularizing. First, the VAE is not a Bayesian model (i.e., we still optimize the likelihood function, not the marginal likelihood). Second, the prior (or rather the marginal distribution over z’s) could be trained and then it serves a different purpose, namely, fitting a model to match the aggregated posterior distribution. Therefore, I find the paragraphs on the first page not fully convincing.
- From the presentation perspective, it is rather a peculiar choice to refer to a figure (Fig. 3) that could be found on page 4. It breaks the flow of reading the paper.
- It is unclear to me why the deterministic variable z is obtained by taking the mean value of y, namely, $z = g_{\psi}(\mu_{\phi}(x))$ (page 4). Shouldn’t it be $z = g_{\psi}(y)$? Otherwise, what’s the point in having a stochastic y?
- Following the previous point, in Eq. 2, it is mentioned that $g_{\psi}(y)$ is used. I start doubting whether the ELBO is correctly calculated.
- I do not see how the proposed idea is different from having a more sophisticated decoder in which we focus on its specific part (i.e., first layers). In other words, I do not see why the proposed analysis in Sect. 4.1 and 4.2 cannot be applied to any decoder.
- The whole paper is written in a pretty confusing manner. One can have an impression that z’s are separate random variables (the authors constantly repeat that the resulting distribution over z’s after applying a mapping g is, e.g., approximately spherical) but they are not used anywhere in the objective function explicitly. I fully understand that it is interesting to see how the inductive biases could be used and that the resulting distributions are of a specific form. However, it does not help a reader to follow the paper. Especially that the stochastic latent variables are y and they are used in the ELBO.
- I tried to go through the code but after spending 30min I gave up. The code is not necessarily helpful to figure out some missing parts of the paper.
- I think the paper presents some interesting ideas, however, it is lost by not necessarily a good way of presenting it.


**Summary Of The Paper:**

The paper proposes to incorporate some inductive biases into VAEs by modifying decoders. The idea is to first transform stochastic latent variables using some specific mapping and then apply a deep neural network. The expected result is that the encoder will be forced to assign z’s in such a way that the mapping can turn them into a manifold of a given form.

**Summary Of The Review:**

The paper shows that modifying a decoder in a VAE could be helpful and allows to still use a simple marginal distribution over z’s (e.g., a standard Gaussian distribution). However, the presentation of the paper is somehow misleading in many places that making it difficult to follow the flow of thoughts. Therefore, I believe the paper is ready to be accepted.

=== UPDATE ===
I would like to thank the authors for their rebuttal. Indeed, some of my concerns were fully addressed and the paper looks better now. However, I am still not fully convinced how to properly choose the transformation $g$ and what is the recipe for that. Anyway, I decided to increase my score to 6.

---

> ### Author Response · Authors · 2021-11-15
> **Reply to Reviewer NaAZ (Part 2)**
>
> >*One can have an impression that $z$’s are separate random variables, but it’s not used in ELBO.*
>
> The representation variable $z=g_\psi(y)$ appears in the ELBO as the input for the decoder.  The density of $z$ does not, because it is defined implicitly and the approach is specifically constructed to avoid it needing to be calculated, while Theorem 1 ensures we still have an implicit valid ELBO defined on the space of $z$ itself.  The fact that the density of the representation variables need not actually appear in the training objective for us to be able to learn them effectively is a crucial aspect of the work.  It is exactly our concept of an intermediary latent space and it ensures the training objective remains easy to estimate and optimize, without requiring any complex non-Gaussian distributions to be learned or matched.
>
> &nbsp;
>
> >*The code is not necessarily helpful to figure out some missing parts of the paper.*
>
> We are currently in the process of adding significant additional documentation to the source code to ensure that it is self-contained.
>
> &nbsp;
>
> >*The presentation of the paper is somehow misleading in many places.*
>
> With the aforementioned corrections, we do not believe the paper is misleading anymore, but highlight that there are simply different interpretations of the approach.  The equivalence between these interpretations is part of the contribution of the work and so it does not make sense to simply omit some of them.  We have, though, tried to make improvements to address your comments by more explicitly delineating these different interpretations, including explicitly laying out the one you have focused on in your review.
>
> We believe the improvements we have to the paper as part of this rebuttal have significantly strengthened it and hope we have addressed the concerns you have with backing its acceptance.  Please let us know if there is anything else you would like further clarification on, or any further improvements you would like to see carried out.

---

> > ### Comment · Reviewer_NaAZ · 2021-11-24
> > **I increase my score to 6**
> >
> > Dear authors,
> >
> > First of all, I would like to thank you for your marvelous rebuttal. Great job!
> > I read your answers very carefully and read the paper again. I must say that now the paper looks indeed better. Therefore, I decided to increase my score to 6. Even though I am leaning towards acceptance, I still don't know how the proposed idea would work in more challenging scenarios (i.e., higher dimensional images or other objects). The necessity of knowing the first layer in the decoder (i.e., the transformation $g$) is somehow limiting in the sense that in many cases this could be highly non-trivial. Nevertheless, I like the idea because of its simplicity and because it doesn't require changing the ELBO at all. BTW, you are right, the ELBO is calculated correctly.
> >
> > Best!

---

> > > ### Author Response · Authors · 2021-11-27
> > > **Re: I increase my score to 6**
> > >
> > > Thank you very much for reconsidering your recommendation and praising the simplicity of our paper!
> > > We agree more complex human knowledge usually requires subtler design for the mapping $g_\psi$. But it would still be much simpler than changing the prior, the encoder(, and perhaps the loss function), which is the current common practice. We think extending the Intel-VAE framework to more complex settings is a very interesting direction for future work, which might serve as a stepping stone to sufficiently combining the advantages of human intelligence and neural networks.

---

> ### Author Response · Authors · 2021-11-15
> **Reply to Reviewer NaAZ (Part 1)**
>
> Thank you for your detailed review and helpful suggestions. We have made a number of updates to the paper to address points raised, while we hope the below helps clear up some points of confusion and potential misunderstandings.
>
> &nbsp;
>
> >*I do not agree [...] that the prior in the VAE framework is about regularizing.  First, the VAE is not a Bayesian model [...] Second, the prior (or rather the marginal distribution over z’s) could be trained and then it serves a different purpose.*
>
> By expressing the ELBO in the form of Eq (1) it is easy to see that the prior can only directly influence the encoder during training.  This influence is through a KL divergence between the two which is why we refer to it as regularizing the encoder.  We note this view has already been presented in many previous works (e.g. [Hoffman & Johnson (2016), Takahashi et al. (2019), and Tomczak & Welling (2018)]) and is not something we claim as a contribution of our own.
>
> We very much agree that the VAE is not really a Bayesian model during training and this is the basis of many of our arguments: specifically, we are arguing that the ‘prior’ is not really a prior in the Bayesian sense and therefore there are better ways to place inductive biases in the model than changing the prior itself.  We have made some edits in the introduction to make this clear.
>
> This viewpoint holds even more strongly once we consider training the prior as this will, by construction, reduce the strength of any inductive biases it contains.  We note that we do consider such prior training by including the Vamp-VAE in the empirical evaluations of Sec.6.
>
> In light of the above, we are a little confused why you see this as a weakness of the work, as your concerns seem based around the viewpoints of previous work, while our own is a clear step away from the interpretations you are concerned about.
>
> &nbsp;
>
> >*From the presentation perspective, it is rather a peculiar choice to refer to a figure (Fig. 3) that could be found on page 4.*
>
> We thank you for this great suggestion. We have duly moved the Figure to the top of page 2 in the revised version.
>
> &nbsp;
>
> >*Why the deterministic variable z is obtained by taking the mean value of y, namely, $z=g_\psi(\mu_\varphi(x))$ (page 4). Shouldn’t it be $z=g_\psi(y)$?*
>
> We believe that there has been a misunderstanding here.  During training (or when otherwise evaluating the ELBO) we indeed take $z=g_{\psi}(y)$.  Here when we refer to using $z=g_\psi(\mu_\varphi(x))$, we are explicitly talking about calculating representations for downstream tasks at test time, for which it is standard to just calculate the encoder mean to produce a deterministic representation, rather than adding the noise to $y$ that is needed for training.
>
> We agree though that this could have been a potential source of confusion and have removed any direct reference to $g_\psi(\mu_\varphi(x))$ in the updated draft to avoid this.
>
> &nbsp;
>
> >*I start doubting whether the ELBO is correctly calculated.*
>
> When calculating the ELBO we are taking $z=g_\psi(y)$.  As we note in the paper, and have made clearer in the update, from the perspective of training this can actually be viewed as the standard ELBO with a particular decoder architecture that uses $g_{\psi}$ as its first layer.  As such, it trivially follows that it is calculated correctly.  We believe it is actually an important contribution of our work that we are able to incorporate inductive biases while still allowing the use of a standard ELBO and Gaussian prior.
>
> &nbsp;
>
> >*How the proposed idea is different from having a more sophisticated decoder in which we focus on its specific part (i.e., first layers)? I do not see why the proposed analysis in Sec. 4.1 and 4.2 cannot be applied to any decoder.*
>
> As we discuss in the general response, the approach can indeed be interpreted, at least from the perspective of training, as a more sophisticated decoder that uses $g_{\psi}$ as its first layer to induce the required inductive biases.  Viewed from this perspective, the key point of difference to the standard VAE setup is in the design of the $g_{\psi}$, which vary substantially from conventional decoder architectures, and the fact that our *representations* are now partial decodings of the latents, such that the decoder becomes part of the representational model itself.  As such, the analysis of Sec.4 can indeed be applied to any decoder.
>
> We have made notable changes to Sec.4 to make this clearer, including adding a new subsection on “Alternative Interpretations”.  We highlight that showing the equivalence between these interpretations is part of the contribution of our work itself.

---

### Official Review · Reviewer_9rm9 · 2021-11-01

**Correctness:** 4
**Technical Novelty And Significance:** 3
**Empirical Novelty And Significance:** 3
**Recommendation:** 6
**Confidence:** 4

**Main Review:**

The technique is reasonable, and the experiments are with a decent comparison with baseline methods. Overall, I think this is solid work. That being said, some reasoning around the proposed framework sounds awkward, due to which I think the technical contribution of the paper is not clearly presented.

My concern is on the novelty of the abstract proposed framework, InteL-VAE. Basically, the transformation of the Gaussian latent variable, $g_\psi$, is just a part of a specific design of the decoder. The authors emphasize that $g_\psi$ is "shared" by both decoder and encoder, and I do understand the intention of such a stance (i.e., the output of $g_\psi$ can be regarded as a transformed latent variable). However, I do not think it makes much sense because in a technical point of view, $g_\psi$ is nothing more than a part of the decoder. Designing a decoder based on some prior knowledge of data is a common practice, and thus the abstract framework of the proposed method can hardly claim significant technical novelty.

I think the technical contribution of the paper rather lies in the particular definitions of $g_\psi$, some of which are presented in the main text, and more are in the appendix. For example, the multi-modality case in section C.2 is very interesting (yet looks ad-hoc). So, I would suggest pivoting the main claim of the paper on these particular instances of $g_\psi$, rather than on the abstract framework of InteL-VAE. For example, it would be great if the multi-modality $g_\psi$ is enough discussed in the main text, rather than being deferred to the appendix.

**Summary Of The Paper:**

The authors propose to introduce some transformation of the Gaussian latent variable of VAEs in order to incorporate specific inductive biases on the semantics of the latent representation. They showcase several examples of such transformation and empirically examine the performance with comparison to baseline methods.

**Summary Of The Review:**

While I think this is solid work, the current presentation is not necessarily kind for readers to understand the technical contribution specific to this paper.

---

> ### Author Response · Authors · 2021-11-15
> **Reply to Reviewer 9rm9**
>
> Thank you for your detailed, helpful, and insightful review, and praise of our work. We have updated the submission file as per your suggestions, incorporating general edits to make the paper ‘kinder’ on the reader, and respond to some of your more specific points below.
>
> &nbsp;
>
> >*$g_\psi$ is just a part of a specific design of the decoder.*
>
> As we explain more fully in the general response, this is a valid and important alternative interpretation of the approach that we agree was not made clear enough in the original submission.  What separates Intel-VAEs from a standard VAE in this interpretation is a) the type of layers we are suggesting using are, as you say, very different from those currently used in practice, and b) that we still use $g_{\psi}$ in the generation of representations, so in this interpretation we have to consider our representation as a partial decoding of the latent.  We have made significant updates to Sec.4 to make this alternative interpretation clear.
>
> Note here that when we talk about sharing of $g_{\psi}$, this is actually not between the encoder and decoder themselves, but between the representational and generative models (which no longer quite match up respectively in our framework).  This viewpoint is important for seeing *why* the Intel-VAE approach is preferable to how things have been done previously, by showing how it can allow inductive biases to be incorporated into the model without undermining the required congruence between the encoder and the prior.  We also believe the conceptual idea that the representational and generative models should not be completely detached if we want to instill inductive biases is an important contribution, and something which would be difficult to convey if we only present $g_{\psi}$ as part of the decoder.
>
> In short, we believe that both interpretations are important and have updated the paper to try and convey both clearly.
>
> &nbsp;
>
> >*I think the technical contribution of the paper rather lies in the particular definitions of $g_{\psi}$ [...] it would be great if [for example] the multi-modality $g_\psi$ is enough discussed in the main text, rather than being deferred to the appendix.*
>
> Thanks for the great suggestion, which was also made by other reviewers and we are delighted to incorporate it. We have followed your advice and moved substantial extra details on the specific $g_{\psi}$ into the main paper and added more emphasis to the importance of these in the narrative.  This includes expanding Sec.6.2 on multi-modality with additional discussion, equations, and figures to ease understanding.  See the general response to all reviewers for more details on the related changes, along with the updated submission file.
>
> &nbsp;
>
> >*A decoder based on some prior knowledge of data is a common practice, and thus the abstract framework of the proposed method can hardly claim significant technical novelty.*
>
> While we agree that our work has novelty in the particular definitions of $g_{\psi}$, which typically instill much stronger and more explicit inductive biases than standard decoders, we do also believe that there is significant novelty in our abstract framework as well.
>
> Firstly, using part of the decoder to construct *representations* with the desired characteristics / inductive biases is itself a novel high-level idea that is highly distinct to the general notion of incorporating prior information into the architecture design.
>
> Secondly, we note that previous papers that have suggested adding comparable inductive biases into VAEs have almost universally done so through direct changes to the prior (see e.g. second paragraph of the introduction); we believe none suggest something comparable to our own framework.  If the approach we suggest was the obvious solution to instilling inductive biases into a VAE a priori, it stands to reason that these papers would have taken our suggested approach instead of an approach we show is significantly inferior. It is certainly the case that we ourselves took a long time to settle on the final suggested approach when trying to tackle this inductive bias problem.

---

> > ### Comment · Reviewer_9rm9 · 2021-11-23
> > **Some more points**
> >
> > Thank you for the authors' rebuttal.
> >
> > I do not disagree that $g_\psi$ of the proposed method can be *interpreted* as something shared between representation and generation, as the authors claim. Yes, it is indeed a valid interpretation. However, my point is that such an interpretation does not necessarily secure the technical novelty of the abstract framework of the proposal. As I mentioned in the initial review, another interpretation that $g_\psi$ is a part of the decoder could explain the limited technical novelty of the abstract methodology, which could not be overruled by other interpretations. That being said, I never mean this work is valueless, hence the suggestion of emphasizing the concrete examples of $g_\psi$ in the paper. I think the updated version is improved in terms of the presentation of the construction of $g_\psi$. My comment on the abstract framework is just that claiming its technical novelty per se would not be so reasonable.
> >
> > > it can allow inductive biases to be incorporated into the model without undermining the required congruence between the encoder and the prior
> >
> > This point of view is interesting. Meanwhile, such a convenience would be at the trade-off with the possibly inappropriate treatment of distribution shape. For example, in [Davidson+ 2018], their $\mathcal{S}$-VAE uses vMF distribution both for posterior and prior, with which appropriate distribution is handled for data on sphere. In contrast, in the authors' method, there is no guarantee that the push-forward measure of $z$ is appropriate because the distribution of $y$ is fixed to be Gaussian. This point could also be discussed more.
> >
> > > Secondly, we note that previous papers that have suggested adding comparable inductive biases into VAEs have almost universally done so through direct changes to the prior (see e.g. second paragraph of the introduction)
> >
> > I would disagree this claim. Many of them incorporate inductive bias via not only prior but also posterior / encoder. For example, [Davidson+ 2018] use vMF distribution for posterior. In [Mathieu+ 2019a], encoder (and also decoder) is designed for distribution on hyperbolic space. The authors' intention of saying so, also in the second paragraph of the introduction, should be elaborated.
> >
> > Given these perspectives, I maintain the initial score, weakly supporting the acceptance.

---

> > > ### Author Response · Authors · 2021-11-24
> > > **Re: Some more points**
> > >
> > > Thank you very much for taking the time to read and respond to our rebuttal, and for the additional points you have raised.  We just wanted to quickly add some clarifications on some of these.
> > >
> > > &nbsp;
> > >
> > > >*Such a convenience would be at the trade-off with the possibly inappropriate treatment of distribution shape [...] in the authors' method, there is no guarantee that the push-forward measure of $z$ is appropriate because the distribution of $y$ is fixed to be Gaussian.*
> > >
> > > We believe there may have been a misunderstanding here as the guarantees about the appropriateness of the distribution on $z$ should be just as strong as previous work: there is no trade-off between the congruence and ensuring we have the desired/appropriate distribution.  Essentially we can think of $y$ as a reparameterization of $z$, and thus we can still express any implied prior (and posterior) over $z$ that we want, with nothing lost in the sharing.  For example, we can recover the uniform distribution on a sphere with a Gaussian prior on $y$ by using exactly the spherical mapping in the paper in the limit $\epsilon \to 0$ (one can experience difficulties in calculating gradients in such limits, but this is an issue that is already experienced by, e.g., [Davidson et al., 2018, Mathieu et al., 2019a]).  This will then ensure that, in the space of $z$, both the prior and the posterior take the desired appropriate form, with Theorem 1 further providing guarantees about the divergence between the two not being worse than that in $y$.
> > >
> > > &nbsp;
> > >
> > > >*I would disagree with this claim (that adding comparable inductive biases into VAEs have almost universally done through direct changes to the prior).*
> > >
> > > Again we feel there may have been a misunderstanding here: the inductive bias mechanism of directly changing both the prior and encoder distributions, as done by e.g. [Davidson et al., 2018, Mathieu et al., 2019a], is exactly the approach our paper is arguing against and that we were suggesting in our rebuttal is what prior work typically uses.  In short, the phrase 'direct changes to the prior' could equally have been 'direct changes to the prior and encoder' in the context of our argument, sorry this was not clear.
> > >
> > > Our key point here was that people have not previously used changes to the decoder as the primary mechanism for simultaneously instilling inductive biases into both the generative models and representations.  We believe that the paper shows quite comprehensively why this is better than instead manually changing both the prior and encoder distributions like [Davidson et al., 2018, Mathieu et al., 2019a].  Moreover, we think that the fact there is still some confusion here actually very much demonstrates the important and non-obvious distinctions our paper is making to previous work.  The original comment in our rebuttal was a response to a suggestion you make that high-level aspects of our approach are common practice, we believe this shows why this is not actually the case.

---

> > > > ### Comment · Reviewer_9rm9 · 2021-11-25
> > > > **Thank you for further response**
> > > >
> > > > Thank you for the further response!
> > > >
> > > > As for the first point, yes I agree that *theoretically*, it would not be impossible to express any prior & posterior over $z$. My intention of having said "trade-off" is that *technically*, it would not be always easy to design $g_\psi$ so that $z$ has a desired distribution shape. For example, I do not think that you can make $z$ to follow vMF distribution by simply transforming $y$ that follows Gaussian, uniform, etc. (for d-dim vMF, to my knowledge, rejection sampling is needed), either in prior or posterior. So, maybe I should have said "desired distribution shape" instead of "appropriate" about $z$'s distribution. I intended to say an obvious thing that nontrivial effort is needed for crafting $g_\psi$, and that the way of crafting $g_\psi$ affects not only the shape of the space of $z$ but also its distribution. I said trade-off because, in the proposed method, the difficulty of designing a good encoder is replaced by the difficulty of designing good $g_\psi$. There is no free lunch, and I think discussing not only why the proposed method is preferable but also why possibly not would be helpful for readers.
> > > >
> > > > As for the second point, my claim is simply summarized what you said:
> > > >
> > > > >  In short, the phrase 'direct changes to the prior' could equally have been 'direct changes to the prior and encoder' in the context of our argument
> > > >
> > > > I just meant this.

---

### Official Review · Reviewer_tE7T · 2021-11-02

**Correctness:** 3
**Technical Novelty And Significance:** 3
**Empirical Novelty And Significance:** 3
**Recommendation:** 8
**Confidence:** 4

**Main Review:**

Firstly, I'd like to thank the authors for an exceptionally well-written paper - it was a joy to read! The well-made figures and plots really help to show the benefits of the proposed model.

The problem of how to effectively introduce biases in VAEs is nicely introduced and discussed. In particular, I found the discussion on the shortcomings of prior modification enlightening and convincing. The proposed model is both simple and intuitive, and I like the connection to normalising flows that the authors make a couple of times, since it gives a nice, mental picture. I would have liked, however, to see some of the equations relating to the inductive biases from the supplementary in the main paper, for instance equations (C.3) through (C.5). I think they are quite important for understanding how the model would work in practice.

I am also not entirely convinced that the transformation should be seen as being shared between the encoder and decoder. I understand why the authors would like to push this view, since the introduced inductive bias, by definition, should improve the learnt representations, which may then be used for downstream tasks. However, my understanding of the model is that the transformation is simply the first layer of the decoder, which then happens to be handcrafted to introduce the appropriate inductive bias. I don't think this view reduces the significance of the proposed model; on the contrary, I find it cleaner to separate the encoder and the decoder this way. (Indeed, some VAE fundamentalists would argue that only the decoder is the model, so decoupling the encoder and the decoder would also please them :) ).
In this light, I don't think theorem 1 is necessary to have in the main paper and, personally, I find the discussion of how to construct inductive biases for different problems both more interesting and practically useful. Of course, if I have misunderstood the model I'm happy to be corrected by the authors.

The experiments are well-thought-out and show increasingly complex aspects of the model - from learning to represent toy data to learning sparse representations of CelebA images. I like the progression of the section, even though the supplementary material is needed to make it completely reproducible.
The point that the authors make that only the topological properties of the data distribution need to be matched by the inductive bias is particularly fascinating. I would very much like to see this discussed further somewhere, though it may be outside the scope of the paper.

I didn't find any technical or mathematical issues in the paper, but there aren't many equations either. This isn't meant as a criticism - I like the simplicity of the model, and it highlights how little is needed to improve the decoder. In that sense, it's a nice reminder of how ineffective neural networks can be.


**Questions**

In the discussion of the links to normalising flows, you mention that a normalising flow would be an unlikely choice for the transformation. I assume that you are referring to the architectural constraints and computational costs of normalising flows, but are there other reasons for not using a normalising flow as the transformation?


**Typos**

* Page 5, fourth paragraph from below: "in start contrast" -> "in stark contrast"
* Page 9, first paragraph from above: "DD significantly degrades generation ~generation~ quality"




**Summary Of The Paper:**

The paper presents a novel method for introducing inductive biases into variational auto-encoders (VAEs). The inductive biases are specified through a deterministic transformation in the latent space, which is shared between the encoder and the decoder. Through experiments on toy data, the authors demonstrate how to design appropriate inductive biases by matching the topological properties of the transformed representations to those of the real data distributions. They further quantify the performance of the model's ability to learn informative representations through experiments on MNIST, Fashion-MNIST, and CelebA, and show that it outperforms competitors both in terms of the generation quality with and without sparsity constraints as well as terms of downstream classification based on the representations.

**Summary Of The Review:**

The proposed model is clearly of interest to the ICLR community. Both the discussion of the issues with modifying the prior and the simple and effective solution of introducing a latent transformation should provoke some interesting discussions and further work. The paper is exceptionally well-written and presented, and while it would be a stretch to call it a groundbreaking paper, it is a solid "accept".

---

> ### Author Response · Authors · 2021-11-15
> **Reply to Reviewer tE7T**
>
> We are grateful for your helpful and insightful review and praise of our work.  We have updated the paper as per your suggestions, and respond to some of your specific comments below.
>
> &nbsp;
>
> >*I would have liked, however, to see some of the equations relating to the inductive biases from the supplementary in the main paper.*
>
> This is an excellent suggestion that was also made by other reviewers.  As explained in our general response to all reviewers, we have moved most of the related material into the main paper.
>
> &nbsp;
>
> >*I am also not entirely convinced that the transformation should be seen as being shared between the encoder and decoder [...] I find it cleaner to separate the encoder and the decoder this way.*
>
> This is another excellent point.  As we explain in detail in the general response, this is a perfectly valid, and quite important, alternative viewpoint with some advantages and some disadvantages compared to how we have chosen to present things.  We have reworked Sec.4 to make all the different interpretations clear, in particular adding an “Alternative Interpretations” subsection.
>
> &nbsp;
>
> >*The point that the authors make that only the topological properties of the data distribution need to be matched by the inductive bias is particularly fascinating. I would very much like to see this discussed further somewhere.*
>
> Thank you for your interest! We have made a small expansion to the related discussion in Sec.6.1 and will look to build on this further as we take the work forward.
>
> &nbsp;
>
> >*Are there other reasons for not using a normalising flow as the transformation?*
>
> As you correctly suggest, there is nothing explicitly wrong with using a normalizing flow here, it is simply that the architecture restrictions would introduce unnecessary constraints and they are often not the easiest class of functions to express inductive biases through.  We have clarified this in the paper.
>
> Thanks for pointing out the typos, these have now been corrected.

---

> > ### Comment · Reviewer_tE7T · 2021-11-22
> > **Acknowledgement of feedback**
> >
> > Thank you for your feedback. And thank you for pointing out that the representation model and the generative model in InteL-VAEs do not correspond directly to the encoder and decoder. This distinction makes it much easier to understand why your model can be seen as being conceptually different from the standard VAE framework.
> >
> > Also, after reading your answers, I think I was too focused on the generative perspective when writing the original review. The representations themselves can indeed be very interesting for interpretability and for this it makes sense to look at the representations after the mapping (i.e., to consider the mapped representations as the inference endpoint rather than as a first step of the decoder).
> >
> > I am happy to see that you were able to fit a discussion of your mappings into the main paper. I think this made your paper even better.
> >
> > Thank you again for a well-written and insightful paper. I will keep my score and recommend acceptance.

---

> > > ### Author Response · Authors · 2021-11-24
> > > **Re: Acknowledgement of feedback**
> > >
> > > Thank you for your time reading and responding to the rebuttal and the new manuscript. We are especially grateful for your appreciation of the insight of our work.

---

### Author Response · Authors · 2021-11-15
**General Response**

We thank all reviewers for their precious time and insightful feedback. We are thrilled that everyone felt the problem we are tackling is important, that all the reviewers thought highly of the specific inductive biases introduced and experiments conducted, that our important conceptual contributions have been appreciated, and that the submission has generally been praised as well-written. We are also pleased to see that all three realizations (multiple-connectivity, multi-modality, and sparsity) of Intel-VAEs have individually won favor with reviewers.

We have taken on board all of your very helpful suggestions and have made noticeable improvements in an updated submission to incorporate these.

Along with responses to individual reviewers, we wanted to respond to two key common points that were raised, both of which we believe we have addressed in our updated submission.

&nbsp;

>*The transformation [should not] be seen as being shared between the encoder and decoder.*

A number of reviewers commented that they were either confused by, or disagree with, the notion that the mapping $g_{\psi}$ is shared between the encoder and decoder.  We believe this concern stems from a combination of a very reasonable potential misunderstanding, that we have now made edits in the paper to avoid, and the presence of multiple valid interpretations of our approach, which we have now more carefully delineated in the draft.

The potential misunderstanding is that though the mapping $g_{\psi}$ is shared between the *representation* and the *generative* models, these do not actually correspond to the encoder and decoder in our framework. As illustrated in Fig.1, and explained in the first two paragraphs of Sec.4, $g_{\psi}$ is regarded as a separate part from the encoder $q_{\phi}$ and decoder $p_{\theta}$. The encoder $q_{\phi}$ composed with $g_{\psi}$ constitute the representation model, while $g_{\psi}$ also combines with the prior and decoder $p_{\theta}$ to form the generative model. The key message is that we should inject our inductive biases directly into both models in a compatible way, and the Intel-VAE allows us to do this.  We have re-written Sec.4 so as to clarify this point (see, in particular, the new second paragraph).

However, we also very much acknowledge that there are different valid interpretations of the Intel-VAE approach. Some reviewers commented that  $g_{\psi}$ is *the first layer of the decoder architecture*.  This is certainly a valid and helpful viewpoint.  What separates Intel-VAEs from a standard VAE in this interpretation is a) the type of layers we are suggesting using a very different from those currently used in practice, and b) that we still use $g_{\psi}$ in the generation of representations, so in this interpretation we have to consider our representation as a partial decoding of the latent.

We believe that both interpretations are important as they each convey their own intuitions and insights, while showing the equivalence between these seemingly distinct views is a contribution of the paper in its own right.  We have therefore reworked Sec.4 to include an explicit “Additional Interpretations” subsection to carefully delineate them, ensure neither is missed, and make clear both are valid.

&nbsp;

>*[The paper should] leave more space for the details of the topological layers used.*

>*I would have liked, however, to see some of the equations relating to the inductive biases from the supplementary in the main paper.*

There was an overwhelming consensus from the reviewers that they would like to see more details in the main paper on the exact inductive biases we consider.  We are more than happy to incorporate these excellent suggestions and are pleased to say that we have managed to find space to provide substantial extra detail on them in the main paper in the update.

To be more specific, we have made the following related improvements:
1. We have introduced mapping designs in more detail in Sec.6, with additional descriptions, formulas, and illustrations.  In particular, we have added all of the key equations to the main paper, along with the new descriptive Fig.5.
2. We have emphasized this perspective in Sec.1.
3. We have also added the appendices themselves to the main pdf, so that anything still not in the main paper can be more easily accessed.

---

### Decision · Program_Chairs · 2022-01-20

**Decision:**

Accept (Poster)

**Comment:**

I am recommending a poster for this paper.  There was considerable discussion and much author response.  The reviews were good (after taking author response and paper revision into account) with one out of three being enthusiastic. There was a concern that the basic idea was technically mis-represented as the inductive bias is being placed in the decoder rather than prior. But I am convinced that it is a reasonable idea to place bias in the decoder and that idea is worth publication.

Personally I think the paper would be much stronger with better empirical evaluation.  I find a focus on MNIST (or fashion MNIST) unconvincing. Results on CelebA should be accompanied by sample image generations.  I would rather see downstream task metrics based on learned features.  This paper cannot be put in the same class as recent results on unsupervised learning of image features for downstream tasks. It remains an open question as to whether this paper provides any contribution in that arena.